# GAPA: Post-Hoc Uncertainty Quantification for Pre-Trained Models via Activation-Space Gaussian Processes

## Abstract

Weight-space uncertainty methods (BNNs, ensembles, Laplace) are difficult to apply post-hoc to frozen foundation models due to retraining requirements or prohibitive second-order computations. We introduce Gaussian Process Activations (GAPA), which replace deterministic activations with Gaussian processes which add principled epistemic uncertainty without altering the original predictions. Using a 1-nearest-neighbour FITC surrogate, GAPA yields closed-form, distance-aware variances with logarithmic time complexity. These variances are propagated through the frozen network using delta method propagation rules. Across regression, classification, segmentation, and language modeling, GAPA matches or exceeds existing methods in calibration and OOD detection while being faster at test time.

## 1 Introduction

While Bayesian methods provide a principled foundation for uncertainty, exact posterior inference in modern neural networks is often intractable (MacKay, 1992). This has motivated decades of *weight-space* approximations: Bayesian neural networks (BNNs) place distributions over parameters but often cannot use an already existing model with deterministic weights and require retraining from scratch (Blundell et al., 2015; Graves, 2011); deep ensembles approximate epistemic uncertainty via multiple independent trainings, multiplying compute and memory (Lakshminarayanan et al., 2017); and Laplace approximations construct Gaussian posteriors around trained weights but depend on expensive second-order curvature estimates—even last-layer Laplace (LL-Laplace) variants become prohibitive for large output dimensions (Daxberger et al., 2021). This creates a practical gap – practitioners need a method that can be applied post-hoc to any pre-trained model without computational overhead or performance degradation (Ovadia et al., 2019). Current methods force an unfeasible choice: either pursue proper uncertainty quantification through expensive approaches, or settle for simple calibration techniques that only adjust confidence without capturing epistemic uncertainty (Guo et al., 2017).

In this work we propose a different path toward uncertainty modeling, focusing on the activation space. Activation space is (i) lower-dimensional than weight space, (ii) semantically meaningful through the backbone's learned representations. We introduce the **Gaussian Process Activation (GAPA)** framework—a drop-in uncertainty layer for frozen pre-trained networks. GAPA replaces a deterministic activation with a Gaussian Process and uses propagation rules to transfer the uncertainty signal through the network. Figure 1 illustrates GAPA's key property: the decision boundary (black line) remains identical to MAP's, while epistemic uncertainty increases with distance from training points (orange/yellow).

**Contributions.**

1. We replace deterministic activations $\phi$ with Gaussian processes whose prior mean equals $\phi$, proving the posterior mean remains $\phi$—thus GAPA outputs a distribution $\mathcal{N}(\phi, \Sigma)$ where the mean exactly preserves the backbone's original predictions and the covariance $\Sigma$ quantifies uncertainty.

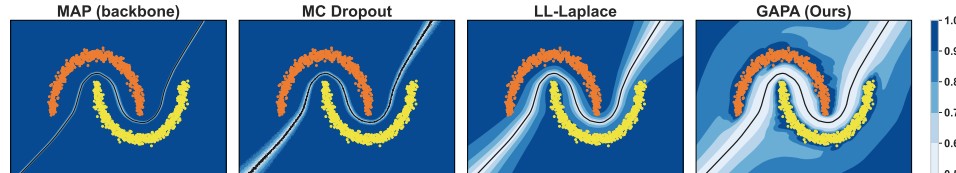

**Figure 1:** Comparison of four uncertainty quantification (UQ) methods on a toy binary task (left→right): MAP (backbone), MC Dropout, LL-Laplace, and GAPA (ours). Background shading indicates (darker = more confident); orange/yellow points are the two classes. GAPA preserves the backbone decision boundary (black).

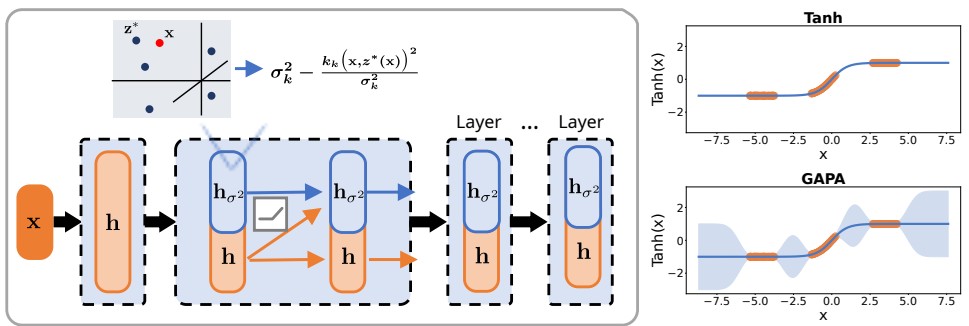

**Figure 2: Left:** The GAPA framework adds a parallel variance path (blue) to frozen networks while preserving mean predictions (orange). The inset shows the 1-NN FITC computation. **Right Top:** Standard tanh activation function. **Right Bottom:** GAPA-augmented tanh with uncertainty bands—the mean (blue line) exactly matches the original tanh while the shaded regions quantify epistemic uncertainty.

2. We develop a 1-nearest-neighbour FITC approximation using training pre-activations as inducing points, achieving $O(\log M)$ retrieval (for $M$ inducing points) and $O(d)$ variance computation (for layer width $d$) with uncertainty.

3. We provide closed-form rules that propagate activation-space (diagonal) covariance to output-space uncertainties via the delta method, and a noisy-input GP (NIGP) corrections for stacked GAPA layers—enabling single-pass inference without sampling or test-time backprop.

4. Experiments across regression, classification, segmentation, and language modeling show GAPA matches or exceeds existing methods while being faster at test time when applied post-hoc to frozen models.

## 2   MODEL PROPOSITION

We begin with a high-level description of the proposed method and then describe each part in detail. The proposed method consists of two key elements: First, GAPA attaches lightweight Gaussian processes to hidden layers that (i) preserve the network's point predictions and (ii) add epistemic uncertainty. The right panels of Fig 2 contrast a standard tanh activation function (top, labelled "Tanh") with its GAPA-augmented counterpart (bottom, labelled "GAPA")—both share identical means, but GAPA adds uncertainty bands (in blue) that grow with distance from training activations (orange points). Second, we utilize mean-preserving uncertainty propagation in the pre-trained, frozen network. In the left panel of Figure 2 we show how GAPA adds a parallel variance path (blue) to the frozen network keeping its original mean path (orange).

The method pipeline is as follows: We first collect activation patterns of a pre-trained model to build a cache of activations on one or more layers. This process only requires forward passes through the network on a dataset we want GAPA to be certain. We then, optionally, do furthest point sampling on the collected activations and set all hyperparameters empirically

from statistics of the data. Starting at the first GAPA layer, we compute its output variance and propagate it forward through subsequent layers. This yields a single-pass, closed-form variance propagation.

## 2.1 Gaussian Process Activation Function

Consider a backbone neural network that has been trained and is now frozen—all weights remain fixed during our uncertainty quantification, with no gradient updates or fine-tuning. Let $d_\ell$ denote the number of neurons at layer $\ell$. The forward pass computes pre-activations

$$\mathbf{z}^\ell = W^\ell \mathbf{h}^{\ell-1} + b^\ell \in \mathbb{R}^{d_\ell}$$

where $W^\ell \in \mathbb{R}^{d_\ell \times d_{\ell-1}}$ and $b^\ell \in \mathbb{R}^{d_\ell}$ are frozen weights and biases, and $\mathbf{h}^{\ell-1} \in \mathbb{R}^{d_{\ell-1}}$ is the previous layer's output. The activation function $\phi^\ell : \mathbb{R}^{d_\ell} \to \mathbb{R}^{d_\ell}$ applies element-wise, i.e., $[\phi^\ell(\mathbf{z}^\ell)]_j = \phi(z_j^\ell)$ for a scalar function $\phi : \mathbb{R} \to \mathbb{R}$ (e.g., ReLU, tanh), producing $\mathbf{h}^\ell = \phi^\ell(\mathbf{z}^\ell)$. GAPA replaces this deterministic activation function with a vector-valued GP:

$$f(\mathbf{z}^\ell) \sim \mathcal{GP}\big(m(\mathbf{z}^\ell), K(\mathbf{z}^\ell, \mathbf{z}'^\ell)\big), \quad f : \mathbb{R}^{d_\ell} \to \mathbb{R}^{d_\ell},$$

with prior mean set to the original activation function $m(\mathbf{z}^\ell) = \phi^\ell(\mathbf{z}^\ell)$. Since the output is $d_\ell$-dimensional, the kernel $K(\mathbf{z}^\ell, \mathbf{z}'^\ell)$ is a $d_\ell \times d_\ell$ covariance matrix. For computational tractability, we use a diagonal structure:

$$K(\mathbf{z}^\ell, \mathbf{z}'^\ell) = \mathrm{diag}(k_1(\mathbf{z}^\ell, \mathbf{z}'^\ell), \ldots, k_{d_\ell}(\mathbf{z}^\ell, \mathbf{z}'^\ell)) \in \mathbb{R}^{d_\ell \times d_\ell},$$

where each $k_j$ is an RBF (squared exponential) kernel $k_j(\mathbf{z}^\ell, \mathbf{z}'^\ell) = c_j^2 \exp\left(-\frac{\|\mathbf{z}^\ell - \mathbf{z}'^\ell\|^2}{2\ell_j^2}\right)$, with neuron-specific prior variance $c_j^2$ and length-scale $\ell_j$.

**Mean preservation.** We pass training data through the frozen network and store the pre-activation values at layer $\ell$. Let $\tilde{Z} = \{\tilde{\mathbf{z}}_1, \ldots, \tilde{\mathbf{z}}_M\}$ be these cached pre-activations, where each $\tilde{\mathbf{z}}_j \in \mathbb{R}^{d_\ell}$ is the pre-activation vector from a training example. For each cached pre-activation $\tilde{\mathbf{z}}_j$, we also store its corresponding activation value $\tilde{\mathbf{h}}_j = \phi^\ell(\tilde{\mathbf{z}}_j) \in \mathbb{R}^{d_\ell}$.

For neuron $i$, the GP posterior mean at test input $\mathbf{z}_*^\ell$ is given by the standard GP formula:

$$\mu_i(\mathbf{z}_*^\ell) = m_i(\mathbf{z}_*^\ell) + \mathbf{k}_i(\mathbf{z}_*^\ell, \tilde{Z})^T \left[\mathbf{K}_i(\tilde{Z}, \tilde{Z}) + \sigma_n^2 \mathbf{I}\right]^{-1} (\tilde{\mathbf{h}}_i - \mathbf{m}_i(\tilde{Z})) \tag{1}$$

where $\mathbf{k}_i(\mathbf{z}_*^\ell, \tilde{Z}) \in \mathbb{R}^M$ is the vector $[k_i(\mathbf{z}_*^\ell, \tilde{\mathbf{z}}_1), \ldots, k_i(\mathbf{z}_*^\ell, \tilde{\mathbf{z}}_M)]^T$ containing kernel evaluations between the test input and cached points; $\mathbf{K}_i(\tilde{Z}, \tilde{Z}) \in \mathbb{R}^{M \times M}$ is the covariance matrix with entries $[\mathbf{K}_i]_{jk} = k_i(\tilde{\mathbf{z}}_j, \tilde{\mathbf{z}}_k)$; $\sigma_n^2$ is a small noise variance for numerical stability (typically $10^{-6}$); $\tilde{\mathbf{h}}_i \in \mathbb{R}^M$ contains the $i$-th neuron's cached activations $[\phi_i^\ell(\tilde{\mathbf{z}}_1), \ldots, \phi_i^\ell(\tilde{\mathbf{z}}_M)]^T$; and $\mathbf{m}_i(\tilde{Z}) \in \mathbb{R}^M$ contains the prior means at cached points, which equals $\tilde{\mathbf{h}}_i$ since our prior mean is the activation function itself.

Since our prior mean equals the activation function ($m_i = \phi_i^\ell$), we have $\tilde{\mathbf{h}}_i = \mathbf{m}_i(\tilde{Z})$, making the correction term zero

$$\mu_i(\mathbf{z}_*^\ell) = \phi_i^\ell(\mathbf{z}_*^\ell) + \mathbf{k}_i(\mathbf{z}_*^\ell, \tilde{Z})^T \left[\mathbf{K}_i + \sigma_n^2 \mathbf{I}\right]^{-1} \underbrace{(\tilde{\mathbf{h}}_i - \tilde{\mathbf{h}}_i)}_{=\,\mathbf{0}} = \phi_i^\ell(\mathbf{z}_*^\ell).$$

Thus the posterior mean is *identical* to the original activation function, perfectly preserving the network's predictions. The posterior variance for neuron $i$ is

$$\sigma_i^2(\mathbf{z}_*^\ell) = k_i(\mathbf{z}_*^\ell, \mathbf{z}_*^\ell) - \mathbf{k}_i(\mathbf{z}_*^\ell, \tilde{Z})^T [\mathbf{K}_i(\tilde{Z}, \tilde{Z}) + \sigma_n^2 \mathbf{I}]^{-1} \mathbf{k}_i(\mathbf{z}_*^\ell, \tilde{Z}), \tag{2}$$

yielding the full predictive covariance for the output of layer $\ell$'s neurons

$$\Sigma^{(\ell)}(\mathbf{z}_*^\ell) = \mathrm{diag}\big(\sigma_1^2(\mathbf{z}_*^\ell), \ldots, \sigma_{d_\ell}^2(\mathbf{z}_*^\ell)\big) \in \mathbb{R}^{d_\ell \times d_\ell}. \tag{3}$$

where $d_\ell$ is the number of neurons at layer $\ell$. This diagonal matrix captures neuron-wise epistemic uncertainty that grows smoothly with distance from the cached pre-activations.

**Why diagonal covariance?** Treating neurons as conditionally independent (diagonal output covariance) is essential for tractability: a full multi-output covariance would require storing and propagating dense $d_\ell \times d_\ell$ matrices across thousands of neurons, which is prohibitive in both memory and compute for modern networks. This diagonal approximation is standard in scalable Bayesian methods and, crucially for our post-hoc setting, sufficient to capture epistemic uncertainty—as we demonstrate empirically in Section 4.

## 2.2 Scalable Inference via 1-NN FITC approximation

Let $N$ be the number of cached pre-activations at layer $\ell$ (obtained by passing the training set once and recording $\mathbf{z}^\ell$). Then, the exact posterior covariance in Equation (3) requires $\mathcal{O}(dN^3)$ computation and $\mathcal{O}(dN^2)$ memory for $d$ neurons and $N$ cached/training points—prohibitive for modern networks. To reduce the cost, we employ two approximations:

**1. FITC approximation** (Snelson and Ghahramani, 2005): Instead of conditioning on all $N$ training points, we use $M \ll N$ inducing points $\tilde{Z} = \{\tilde{\mathbf{z}}_1, \ldots, \tilde{\mathbf{z}}_M\}$, reducing complexity to $\mathcal{O}(dM^3)$.

**2. 1-Nearest Neighbor selection**: We set $M = 1$ dynamically by selecting the nearest cached pre-activation for each test input $\mathbf{z}^*(\mathbf{z}^\ell) = \arg\min_{\tilde{\mathbf{z}} \in \tilde{Z}} \|\mathbf{z}^\ell - \tilde{\mathbf{z}}\|_2$, retrieved via FAISS (Douze et al., 2024) in $\mathcal{O}(\log M)$ time.

Under these approximations, the posterior mean for the output of neuron $i$ remains $\mu_i(\mathbf{z}^\ell) = \phi_i^\ell(\mathbf{z}^\ell)$ exactly, while its posterior variance simplifies to

$$\sigma_i^2(\mathbf{z}^\ell) = c_i^2 \left( 1 - \exp\left( -\frac{\|\mathbf{z}^\ell - \mathbf{z}^*\|^2}{2\ell_j^2} \right) \right). \tag{4}$$

This closed-form expression shows that variance grows smoothly with distance $\|\mathbf{z}^\ell - \mathbf{z}^*\|$ from the nearest cached activation—see the right-most plot in Figure 6 where uncertainties increase as we move away from training data.

**Computational complexity.** We use FAISS (Douze et al., 2024), a library for efficient similarity search, to index the $M$ inducing points for fast nearest-neighbor retrieval. Building this index requires $\mathcal{O}(Md_\ell)$ one-time setup. Per-query inference costs only $\mathcal{O}(\log M + d_\ell)$—the nearest neighbor search plus $d_\ell$ scalar variance computations. The full derivation of the 1-NN FITC variance formula in Equation (4) is provided in Appendix E.

## 2.3 Variance Propagation Through The Networks

To obtain output-space uncertainty from GAPA layers placed within the network, we propagate variances forward through subsequent layers. We maintain diagonal covariance matrices throughout: $\Sigma_\mathbf{h}$ denotes a variance vector with the same dimensionality as the layer output vector $\mathbf{h}$. Recall that each GAPA layer outputs a Gaussian

$$\mathbf{h}^\ell \mid \mathbf{z}^\ell \sim \mathcal{N}(\boldsymbol{\mu}^\ell(\mathbf{z}^\ell), \Sigma^{(\ell)}(\mathbf{z}^\ell)), \qquad \boldsymbol{\mu}^\ell(\mathbf{z}^\ell) = \boldsymbol{\phi}^\ell(\mathbf{z}^\ell),$$

where the mean equals the original activation function.

We consider three propagation scenarios—(i) linear layers, (ii) nonlinear activations, and (iii) stacked GAPA layers. Specialized architectures (self-attention, LayerNorm) are derived in Appendix K.

**(i) Linear Transformation of Variance.** For a linear transformation $\mathbf{z} = W\mathbf{h} + b$ where $\mathbf{h}$ has diagonal covariance $\Sigma_\mathbf{h}$ (i.e., components of $\mathbf{h}$ are independent), the output covariance $\Sigma_\mathbf{z}$ remains diagonal with entries $[\Sigma_\mathbf{z}]_i = \sum_j W_{ij}^2 [\Sigma_\mathbf{h}]_j$. This follows from the independence assumption $\mathrm{Var}(z_i) = \mathrm{Var}(\sum_j W_{ij} h_j) = \sum_j W_{ij}^2 \mathrm{Var}(h_j)$.

**(ii) Propagation Rules for Non-Linear Activations.** For a non-linear activation $y = g(z)$ applied to a Gaussian random variable $z \sim \mathcal{N}(\mu, \sigma^2)$, we approximate $g(z)$ by a

first-order Taylor expansion around $\mu$: $g(z) \approx g(\mu) + g'(\mu)(z - \mu)$. Under this approximation, $y$ is approximately Gaussian with mean $\mathbb{E}[y] \approx g(\mu)$ and variance $\mathrm{Var}(y) \approx (g'(\mu))^2 \sigma^2$, since $z - \mu \sim \mathcal{N}(0, \sigma^2)$.

**(iii) Stacking GAPA layers (noisy-input GP).** With multiple GAPA layers, each layer passes forward *mean and variance vectors* rather than deterministic activations. Because GAPA is mean-preserving, the predictive mean at the next GAPA layer does *not* change under input uncertainty ($\mu_i(\mathbf{z}^\ell) = \phi_i^\ell(\mathbf{z}^\ell)$ for all $i$). However, the predictive *variance* must account for uncertain inputs. Following the noisy-input GP (NIGP) correction (McHutchon and Rasmussen, 2011), we add a term to the epistemic variance. Let $\Sigma_{\mathbf{z}}$ denote the (diagonal) input covariance entering the current GAPA layer. Then for neuron $i$, $\lambda_i(\mathbf{z}^\ell) = \left(\nabla_{\mathbf{z}^\ell} \mu_i(\mathbf{z}^\ell)\right)^\top \mathrm{diag}(\Sigma_{\mathbf{z}}) \left(\nabla_{\mathbf{z}^\ell} \mu_i(\mathbf{z}^\ell)\right)$, and the total predictive variance becomes

$$
\sigma_i^2(\mathbf{z}^\ell) \;=\; \underbrace{c_i^2 \;-\; \frac{k_i(\mathbf{z}^\ell, \mathbf{z}^*)^2}{c_i^2 + \sigma_n^2}}_{\text{epistemic (1-NN FITC)}} \;+\; \underbrace{\lambda_i(\mathbf{z}^\ell)}_{\text{input-uncertainty (NIGP)}} \;+\; \underbrace{\sigma_{y,i}^2}_{\text{aleatoric noise}} \;.
$$

In our elementwise setting, $\mu_i(\mathbf{z}^\ell) = \phi_i^\ell(z_i^\ell)$ and $\lambda_i(\mathbf{z}^\ell) = (\phi_i'(z_i^\ell))^2 [\Sigma_{\mathbf{z}}]_i$. Here, $\sigma_{y,i}^2$ models observation (aleatoric) noise; we set $\sigma_{y,i}^2 = 0$ for classification and either learn a heteroscedastic head or calibrate a scalar in regression (Sec. 2.4). The mean prediction remains $\mu_i(\mathbf{z}^\ell) = \phi_i^\ell(\mathbf{z}^\ell)$ regardless of input uncertainty; only the variance is updated by the NIGP correction.

### 2.4 Hyperparameter Strategy

We do not optimize GP kernel hyperparameters (or the inducing set) via gradient descent or marginal likelihood. Instead, we set them once from cached training statistics: the RBF amplitude $c_i^2$ to the empirical variance of neuron $i$'s activations, the length–scale $\ell_i$ to the median pairwise distance between cached pre-activations, the inducing set $\tilde{Z}$ via farthest-point sampling, and a small jitter $\sigma_n^2$ (e.g., $10^{-6}$) for numerical stability (details in Appendix F.1). This is necessary because our GP prior mean equals the activation ($m_i = \phi_i$), so marginal-likelihood optimisation admits a degenerate solution $c_i^2 \to 0$ that collapses epistemic uncertainty.

For classification, we keep GAPA kernel hyperparameters fixed and do not learn any additional parameters. We compute predictive probabilities using the Laplace-bridge approximation, which adjusts each logit by its uncertainty before applying softmax—effectively downweighting uncertain predictions (see Appendix I for the exact formula). We do not add a separate aleatoric noise term since the softmax output already captures class uncertainty through its probability distribution.

For regression, we keep the backbone and GAPA kernels fixed but learn an input-dependent aleatoric noise term. We train a small MLP head to predict heteroscedastic noise $\sigma_{\mathrm{ale}}^2(\mathbf{x})$, which is added to the epistemic variance $\sigma_{\mathrm{epi}}^2(\mathbf{x})$ from GAPA. The total variance $\sigma_{\mathrm{tot}}^2 = \sigma_{\mathrm{epi}}^2 + \sigma_{\mathrm{ale}}^2$ is used in a Gaussian negative log-likelihood loss. Since only the noise head is trained while the backbone remains frozen, mean predictions are preserved exactly. See Appendix F.2 for implementation details.

## 3 Related Work

Post-hoc uncertainty for frozen networks has been approached along several lines. *Feature-based* and output-layer methods analyze or modify representations near the head: ODIN/Mahalanobis-style detectors (Hendrycks and Gimpel, 2016; Lee et al., 2018; Postels et al., 2020), distance-aware heads such as DUQ (Van Amersfoort et al., 2020) and SNGP (Liu et al., 2020). *Calibration* techniques (e.g., temperature scaling and Platt scaling) adjust confidence without modeling epistemic uncertainty (Guo et al., 2017; Platt et al., 1999); *conformal prediction* provides coverage guarantees but does not by itself supply a distance-aware epistemic signal (Vovk et al., 2005; Shafer and Vovk, 2008; Angelopoulos and Bates,

2021). *Sampling-based* approaches include MC Dropout (Gal and Ghahramani, 2016), which requires multiple stochastic forward passes, and Deep Ensembles (Lakshminarayanan et al., 2017), which multiply training and inference costs. GP-style probes on penultimate features (e.g., Linear Probing, SNGP) place uncertainty primarily at the output layer (Liu et al., 2020).

Last-Layer Laplace (LL-Laplace) and variants (KFAC, ELLA, VaLLA) construct Gaussian posteriors over the final linear head and enable post-hoc attachment to frozen backbones (Daxberger et al., 2021; Ortega et al., 2023). However, scalability is limited for high-dimensional outputs: computing per-class logit variances $x_*^\top \Sigma_w x_*$ scales as $O(Vd)$ per token for vocabulary size $V$ and hidden width $d$, which becomes prohibitive for large vocabularies ($V \gtrsim 50k$), even under diagonal approximations. As a result, LL-Laplace is typically used for small downstream heads (classification, reward models) rather than full next-token prediction.

## 4 Results

In this section, we demonstrate GAPA's effectiveness and broad applicability across diverse tasks and model families. We present results on standard regression benchmarks, classification, and language models, with additional evaluations on ResNets (Appendix A), image segmentation (Appendix B), and LLaMA-3.2 (Appendix C) in the appendix. Ablation studies examining GAPA layer placement, number of inducing points, and sampling strategies are provided in Appendix M.

### 4.1 Regression

**Table 1:** Results on regression datasets. Best values are in purple, and second-best in teal. An asterisk (*) indicates a last-layer LLA variant. Results are averages over 5 random seeds; standard deviations ($< 10^{-3}$ in all cases) are omitted for brevity. The full table with stds can be found in Table 6 in the Appendix.

| Model | Airline | | | Year | | | Taxi | | |
|---|---|---|---|---|---|---|---|---|---|
| | **NLL** | **CRPS** | **CQM** | **NLL** | **CRPS** | **CQM** | **NLL** | **CRPS** | **CQM** |
| MAP (backbone) | 5.121 | 18.695 | 0.148 | 3.673 | 5.023 | 0.134 | 3.775 | 3.755 | 0.211 |
| LLA Diag | 5.125 | 18.648 | 0.143 | 3.647 | 4.917 | 0.088 | 3.722 | 3.990 | 0.257 |
| LLA KFAC | 5.127 | 18.631 | 0.142 | 3.648 | 4.915 | 0.086 | 3.706 | 3.986 | 0.256 |
| LLA* | 5.127 | 18.631 | 0.141 | 3.648 | 4.915 | 0.086 | 3.726 | 3.985 | 0.256 |
| LLA* KFAC | 5.127 | 18.631 | 0.141 | 3.648 | 4.914 | 0.086 | 3.726 | 3.985 | 0.256 |
| ELLA | 5.388 | 21.671 | 0.413 | 4.020 | 6.049 | 0.424 | 3.885 | 3.680 | 0.219 |
| VaLLA 100 | 4.963 | 18.814 | 0.099 | 3.515 | 5.004 | 0.047 | 3.235 | 3.999 | 0.149 |
| VaLLA 200 | 4.965 | 18.788 | 0.098 | 3.485 | 4.970 | 0.041 | 3.232 | 3.979 | 0.142 |
| Dropout | 5.102 | 19.066 | 0.938 | 3.689 | 5.128 | 0.939 | 3.849 | 4.592 | 0.951 |
| Ensemble | 5.053 | 18.205 | 0.933 | 3.639 | 4.833 | 0.938 | 3.631 | 3.384 | 0.961 |
| **GAPA (ours)** | 4.946 | 18.068 | 0.103 | 3.470 | 4.663 | 0.014 | 3.112 | 4.035 | 0.104 |

We compare GAPA against state-of-the-art post-hoc Laplace-based methods (VaLLA, LLA variants, ELLA (Daxberger et al., 2021; Izmailov et al., 2020; Ortega et al., 2023)) on three benchmarks: UCI Year, Airline (Dutordoir et al., 2020), and Taxi (Salimbeni and Deisenroth, 2017), using original train/test splits. Performance is evaluated using Negative Log-Likelihood (NLL), Continuous Ranked Probability Score (CRPS), and Centered Quantile Metric (CQM), with detailed definitions in Appendix J.1. Table 1 shows GAPA achieves best performance across nearly all metrics, with only two exceptions: third for CRPS on Taxi and marginally higher CQM than VaLLA on Airline.

### 4.2 Classification

We evaluate GAPA's performance on classification tasks by assessing predictive accuracy, calibration, and out-of-distribution (OOD) detection capabilities. Key metrics include Accuracy (ACC), Negative Log-Likelihood (NLL), Expected Calibration Error (ECE), and the Area Under the ROC Curve (AUC) for OOD detection using predictive entropy (OOD-Entropy) and the BALD score (OOD-BALD). Detailed definitions of these metrics are

provided in Appendix J.2. The ResNet and image segmentation experiment can be found in Appendix A and B respectively.

**Table 2:** Results on classification datasets. Best values are in purple, second-best in teal. Results are averages over 5 random seeds; standard deviations ($< 10^{-3}$ in all cases) are omitted for brevity. The full version with standard deviations can be found in Table 7 in the Appendix.

| Model | MNIST | | | | | FMNIST | | | | |
|---|---|---|---|---|---|---|---|---|---|---|
| | ACC | NLL | ECE | OOD | BALD | ACC | NLL | ECE | OOD | BALD |
| MAP (backbone) | 0.978 | 0.068 | 0.005 | 0.919 | 0.919 | 0.859 | 0.392 | 0.007 | 0.846 | 0.821 |
| LLA Diag | 0.976 | 0.177 | 0.105 | 0.932 | 0.941 | 0.856 | 0.421 | 0.057 | 0.872 | 0.873 |
| LLA KFAC | 0.978 | 0.102 | 0.042 | 0.971 | 0.971 | 0.858 | 0.395 | 0.020 | 0.909 | 0.970 |
| LLA* | 0.978 | 0.070 | 0.009 | 0.924 | 0.924 | 0.859 | 0.395 | 0.019 | 0.850 | 0.716 |
| LLA* KFAC | 0.979 | 0.070 | 0.009 | 0.923 | 0.928 | 0.859 | 0.394 | 0.017 | 0.849 | 0.717 |
| ELLA | 0.978 | 0.068 | 0.005 | 0.919 | 0.912 | 0.859 | 0.392 | 0.007 | 0.846 | 0.765 |
| VaLLA 100 | 0.978 | 0.068 | 0.005 | 0.919 | 0.934 | 0.865 | 0.382 | 0.019 | 0.925 | 0.963 |
| VaLLA 200 | 0.978 | 0.068 | 0.005 | 0.919 | 0.934 | 0.867 | 0.378 | 0.020 | 0.937 | 0.970 |
| Linear Probing | 0.977 | 0.117 | 0.015 | 0.884 | 0.883 | 0.858 | 0.395 | 0.048 | 0.785 | 0.776 |
| GPP | 0.978 | 1.648 | 0.784 | 0.934 | 0.904 | 0.857 | 1.716 | 0.692 | 0.867 | 0.962 |
| Dropout | 0.978 | 0.072 | 0.009 | 0.923 | 0.944 | 0.858 | 0.393 | 0.009 | 0.850 | 0.911 |
| Ensemble | 0.979 | 0.069 | 0.038 | 0.936 | 0.962 | 0.859 | 0.373 | 0.041 | 0.863 | 0.938 |
| DDU | 0.978 | 0.068 | 0.005 | 0.921 | 0.919 | 0.859 | 0.392 | 0.007 | 0.876 | 0.983 |
| **GAPA k=1** | 0.978 | 0.109 | 0.049 | 0.960 | 0.972 | 0.859 | 0.389 | 0.013 | 0.973 | 0.993 |
| **GAPA k=50** | 0.978 | 0.080 | 0.023 | 0.962 | 0.976 | 0.859 | 0.388 | 0.011 | 0.944 | 0.993 |
| **GAPA k=500** | 0.978 | 0.073 | 0.016 | 0.963 | 0.976 | 0.859 | 0.390 | 0.009 | 0.920 | 0.993 |

| Model | MNIST | | FMNIST | |
|---|---|---|---|---|
| | Train | Test | Train | Test |
| LLA Diag | 2.34K | 230.9 | 2.34K | 221.9 |
| LLA KFAC | 130.0 | 1.85K | 129.9 | 1.84K |
| LLA* | 24.0K | 64.3 | 24.0K | 64.3 |
| LLA* KFAC | 31.2 | 17.6 | 31.2 | 17.4 |
| ELLA | 821.8 | 148.7 | 827.1 | 149.9 |
| VaLLA 100 | 2.19K | 16.4 | 495.1 | 16.4 |
| VaLLA 200 | 3.43K | 18.3 | 767.7 | 19.3 |
| Linear Probing | 2.78K | 3.6 | 2.64K | 3.8 |
| GPP | 5.79K | 23.5K | 5.57K | 2.27K |
| Dropout | — | 4.3 | — | 4.3 |
| Ensemble | — | 11.9 | — | 11.9 |
| **GAPA** | 91.9 | 2.1 | 92.1 | 2.1 |
| **GAPA k=50** | 91.9 | 4.0 | 92.1 | 4.0 |
| **GAPA k=500** | 91.9 | 16.3 | 92.1 | 16.3 |

**Figure 3: Left:** Training and test times on MNIST and FMNIST for various models (in seconds; K = ×1000). **Right:** Predictive NLL under rotation corruption for MNIST (left sub-panel of the plot) and FMNIST (right sub-panel of the plot); lower NLL is better. All reported results are averages over 5 random seeds; standard deviations (in all cases $< 10^{-3}$) are omitted for brevity.

We evaluate GAPA on MNIST (LeCun et al., 1998) and Fashion-MNIST (Xiao et al., 2017) using a 2-layer fully connected network (200 units per layer, tanh activations) following Ortega et al. (2023). GAPA layers are applied post-hoc to each activation and pre-softmax logits. For OOD detection, MNIST and FMNIST serve as reciprocal OOD datasets.

Table 2 and Figure 3 show that while GAPA has higher NLL than some baselines, it excels at uncertainty quantification: achieving the highest OOD-AUC (0.960/0.973) and OOD-BALD (0.972/0.993) across all methods. Inference takes only 2.05s—orders of magnitude faster than Laplace variants. Under rotation corruption (Figure 3, right), GAPA maintains the lowest NLL across all angles, correctly identifying OOD inputs with appropriate uncertainty inflation.

### 4.3 LANGUAGE MODELS

We evaluate GAPA on two GPT-style language models: TinyStories (60M) (Eldan and Li, 2023) and GPT-2 Small (124M) (Radford et al., 2019). In Appendix C we provide additional experiments for LLaMA-3.2-3B.

For both models, we log 1.5M embedding vectors $h(t) \in \mathbb{R}^{768}$ from random training sequences and time steps at multiple layers. This takes $\approx$1–2 hours on a consumer GPU and yields a hard-drive footprint of $\approx$5 GB per layer. We then apply furthest-point sampling to subsample the activation logs; we ablate the effect of the inducing set size $N_{\text{inducing}}$ in the results.

We consider a token-level corruption detection task. Given an in-distribution sequence, we replace an $\epsilon$-fraction of tokens with draws from the vocabulary distribution [1]. For each position $t$ in a (possibly corrupted) sequence $x$, we derive an uncertainty score $u(t)$ from the model's next-token distribution $p(\cdot \mid x_{\leq t})$. Corruption detection is then framed as a token-level binary classification task: predict whether $x(t)$ was replaced, using $u(t)$ as the signal. We report AUROC across different noise levels.

**Post-hoc uncertainty scores**   Let $\boldsymbol{\ell}(t) \in \mathbb{R}^V$ be the logits at time $t$ and $\boldsymbol{\ell}_{\sigma^2}(t) \in \mathbb{R}^V$ the corresponding predictive logit variances from GAPA. Define

$$p = \text{softmax}(\boldsymbol{\ell}), \qquad \tilde{p} = \text{softmax}\Big(\boldsymbol{\ell}/\sqrt{1 + (\pi/8)\,\boldsymbol{\ell}_{\sigma^2}}\Big),$$

where $\tilde{p}$ uses the Laplace bridge approximation (Bishop and Nasrabadi, 2006). We report (i) aleatoric AU $= -\sum_v p \log p$, (ii) total TU $= -\sum_v \tilde{p} \log \tilde{p}$, (iii) epistemic EU $=$ TU $-$ AU uncertainty, and (iv) MSP $- \max_v \ell_v$. We additionally include a temperature-scaling oracle that searches $\tau$ to maximize test AUROC using $-\sum_{v=1}^{V} \text{softmax}(\boldsymbol{\ell}/\tau)_v \log \text{softmax}(\boldsymbol{\ell}/\tau)_v$. This is *not* a fair baseline (it tunes on the test metric) but provides an empirical upper bound for methods that only apply a global rescaling of logits. We denote as 'Ours' both TU and EU as they utilize GAPA for epistemic uncertainty quantification.

Our focus is strictly training-free and tuning-free uncertainty from a frozen model. Unlike methods that modify weights or require validation-set fitting (e.g., QLoRA-based posteriors, layer-Laplace, ensembles), GAPA requires no gradients, no weight updates, and no model changes. We therefore compare against baselines that can be computed directly from the pre-trained model's outputs. Unless stated otherwise, we attach GAPA to four layers without tuning: after positional embeddings (*base*), and after the first, a middle, and the last transformer block (GPT-2: [base, 0, 5, 11]; TinyStories: [base, 0, 1, 5]).

**Results**   In Table 3a and 3b we show the results for GAPA with 4 layers. We observe that in both cases the GAPA augmented predictive uncertainties outperform both baselines and (especially for GPT-2) the optimal temperature-scaled logit approach. The difference in performance becomes more prominent for larger noise levels. Interestingly, for GPT-2 the epistemic uncertainty alone, is not a strong predictor for token corruption; only combining both sources of uncertainty via TU($\boldsymbol{\ell}, \boldsymbol{\ell}_{\sigma^2}$) leads to strong performance. We believe its likely that for GPT-2, or large language models in general, approximating the full data distribution with a reasonable amount of inducing points is infeasible by itself, however it does add useful information that can be intertwined with the entropy of the logits. For the TinyStories model, EU alone is a strong predictor for corruption, and so is TU, here it appears that the inducing point can form an accurate data distribution.

Figure 4 (top row) shows the performance as a function of the inducing points. Overall we observe that performance continuously rises as we increase the number of points for GPT-2, whereas for TinyStories a saturation in performance is achieved for around $10{,}000$ inducing points. Overall, we observe that as we increase the number of inducing points GAPA will eventually outperform even the post-hoc temperature tuned baseline. Next, we investigate the effect of the layer position and amount of GAPA layers in Figure 4 (bottom). We see that for GPT-2 the best performance is achieved around the second to last layer, or for a combination. Single GAPA layers at early layers perform worse. For TinyStories the performance is more uniform, where last and first layer GAPAs alone perform slightly worse. In Appendix D we show example visualizations of the predicted uncertainty values. We can see that both *AU* and *EU* spike at distinct positions and unexpected turns seem to correlate with a spike in epistemic uncertainty.

---

[1] For GPT-2 we use a uniform distribution, because TinyStories has a limited vocabulary, we first estimate token frequency and then use a power-law sampling scheme with $\alpha = 1.2$ over the top-10k GPT-2 tokens.

**Table 3:** Token–corruption detection AUROC (↑) at different noise levels $\epsilon$, for GAPA architecture and $N_{\text{inducing}} = 7500$ inducing points. Metrics are averaged over 5 evaluation seeds with $168 \times 16$ batches and sequences of length 750 each.

**(a)** GPT-2 Small (123 Mio., [base, 0, 5, 11]).

| $\epsilon$ | $\text{EU}(\ell, \ell_{\sigma^2})$ (Ours) | $\text{TU}(\ell, \ell_{\sigma^2})$ (Ours) | $\text{AU}(\ell)$ | $-\max_v \ell$ | $H(\ell/\tau_{\text{opt.}})$ | $1/\tau_{\text{opt.}}$ |
|---|---|---|---|---|---|---|
| 0.005 | 0.31±0.001 | 0.87±0.000 | 0.70±0.001 | 0.65±0.001 | 0.84±0.001 | 0.27 |
| 0.010 | 0.30±0.001 | 0.87±0.001 | 0.70±0.001 | 0.65±0.001 | 0.84±0.001 | 0.27 |
| 0.100 | 0.29±0.001 | 0.81±0.000 | 0.72±0.001 | 0.67±0.001 | 0.78±0.001 | 0.40 |
| 0.200 | 0.33±0.000 | 0.77±0.000 | 0.68±0.000 | 0.65±0.000 | 0.71±0.000 | 0.27 |
| 0.300 | 0.37±0.000 | 0.73±0.000 | 0.63±0.000 | 0.61±0.001 | 0.67±0.000 | 0.12 |

**(b)** TinyStories (60 Mio., [base, 0, 1, 5]).

| $\epsilon$ | $\text{EU}(\ell, \ell_{\sigma^2})$ (Ours) | $\text{TU}(\ell, \ell_{\sigma^2})$ (Ours) | $\text{AU}(\ell)$ | $-\max_v \ell$ | $H(\ell/\tau_{\text{opt.}})$ | $1/\tau_{\text{opt.}}$ |
|---|---|---|---|---|---|---|
| 0.005 | 0.80±0.001 | 0.76±0.001 | 0.69±0.001 | 0.65±0.001 | 0.79±0.001 | 0.40 |
| 0.010 | 0.81±0.000 | 0.76±0.000 | 0.69±0.000 | 0.65±0.001 | 0.79±0.001 | 0.40 |
| 0.100 | 0.77±0.000 | 0.74±0.000 | 0.67±0.000 | 0.64±0.000 | 0.76±0.000 | 0.40 |
| 0.200 | 0.73±0.000 | 0.71±0.000 | 0.65±0.000 | 0.62±0.000 | 0.72±0.000 | 0.40 |
| 0.300 | 0.69±0.000 | 0.67±0.000 | 0.63±0.000 | 0.60±0.000 | 0.68±0.000 | 0.40 |

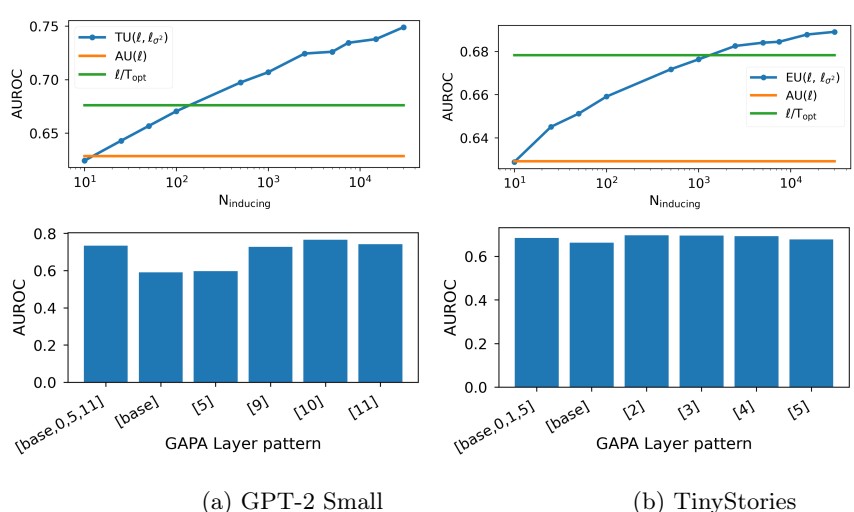

(a) GPT-2 Small        (b) TinyStories

**Figure 4:** Ablation studies: Effect of the number of inducing points $N_{\text{inducing}}$ (top), effect of GAPA layer positions (bottom) on token–corruption detection AUC for $\epsilon = 0.3$. (a) For GPT-2 Small we plot the AUC using TU (blue) with GAPA at layers [base, 0, 5, 11]. (b) For TinyStories we plot the AUC using EU (blue) with GAPA at layers [base, 0, 1, 5]. In the top plots we also show the $\ell/T_{\text{opt}}$ bound (green) as an upper threshold of what can be achieved by global logits scaling.

.

## 5 Conclusion and Limitations

In this work, we introduced GAPA, a novel post-hoc framework that quantifies uncertainty in pre-trained neural networks by modeling activation-space uncertainty with a scalable (FITC with closest nearest neighbor) multi-input, multi-output Gaussian Process, while preserving the base model's mean predictions and propagating uncertainties via delta approximation. Extensive empirical validation across diverse tasks, including regression, classification, high-dimensional image segmentation, and large language models like GPT-2, demonstrated GAPA's broad applicability and competitive performance against Laplace approximations and other baselines. While GAPA offers computational advantages, achieving strong performance on high-dimensional models requires large numbers of inducing points. Future work should develop hierarchical selection schemes (e.g., k-means/IVF clustering with

local 1-NN) where the inducing points are dynamically selected from the nearest clusters. This could dramatically improve the inducing point efficiency while maintaining performance.

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

# A   ResNets Pretrained Neural Networks

**Table 4:** GAPA and baselines on CIFAR-10 with ResNet backbones. Results are averages over 5 random seeds; standard deviations ($< 10^{-3}$ universally) are omitted for brevity.

| | ResNet-20 | | | ResNet-32 | | | ResNet-44 | | | ResNet-56 | | |
| --- | --- | --- | --- | --- | --- | --- | --- | --- | --- | --- | --- | --- |
| | ACC | NLL | OOD | ACC | NLL | OOD | ACC | NLL | OOD | ACC | NLL | OOD |
| MAP | 92.6 | 0.282 | 0.876 | **93.5** | 0.292 | 0.909 | **94.0** | 0.275 | 0.885 | **94.4** | 0.252 | 0.924 |
| MF-VI | **92.7** | **0.231** | – | **93.5** | 0.222 | – | 93.9 | 0.206 | – | **94.4** | 0.188 | – |
| SNGP | 92.4 | 0.266 | – | 93.2 | 0.256 | – | 93.8 | 0.242 | – | 93.8 | 0.229 | – |
| GP (subset) | 92.6 | 0.555 | – | 93.4 | 0.462 | – | 93.6 | 0.424 | – | **94.4** | 0.403 | – |
| LLA Diag | 92.6 | 0.260 | 0.866 | **93.5** | 0.242 | 0.882 | **94.0** | 0.218 | 0.860 | 94.3 | 0.195 | 0.923 |
| LLA KFAC | 92.6 | 0.241 | 0.877 | **93.5** | 0.229 | 0.903 | **94.0** | 0.213 | 0.855 | **94.4** | 0.193 | 0.917 |
| LLA* | 92.6 | 0.269 | – | **93.5** | 0.259 | – | **94.0** | 0.237 | – | **94.4** | 0.213 | – |
| LLA* KFAC | 92.6 | 0.271 | OOM | **93.5** | 0.260 | OOM | **94.0** | 0.232 | OOM | **94.4** | 0.202 | OOM |
| ELLA | 92.5 | 0.233 | OOM | **93.5** | 0.215 | OOM | 93.9 | 0.204 | OOM | **94.4** | 0.187 | OOM |
| Sampled LLA | 92.5 | **0.231** | - - | **93.5** | 0.217 | – | **94.0** | 0.200 | – | **94.4** | 0.185 | – |
| VaLLA | 92.4 | **0.231** | **0.940** | 93.2 | **0.212** | **0.933** | 93.8 | 0.201 | **0.928** | 94.2 | 0.188 | **0.960** |
| **GAPA (ours)** | 92.6 | 0.258 | 0.907 | **93.5** | 0.259 | 0.926 | **94.0** | 0.230 | 0.903 | **94.4** | 0.230 | 0.935 |

We further evaluate GAPA on the CIFAR-10 dataset (Krizhevsky et al., 2009) using pre-trained ResNet architectures of varying depths (ResNet-20, -32, -44, and -56) (He et al., 2016). Given the potentially high dimensionality of intermediate feature spaces in these deeper models, GAPA was applied post-hoc to the pre-activations of the final fully connected layer. For out-of-distribution (OOD) detection in this setup, we used the SVHN dataset (Netzer et al., 2011) as the OOD benchmark against CIFAR-10 as the in-distribution data. Table 4 reports results on CIFAR using ResNet-20, 32, 44, and 56 backbones. In terms of accuracy, GAPA matches or slightly lags the best baselines: it achieves 92.6% on ResNet-20, 93.5% on ResNet-32, and 94.0–94.4% on the deeper variants, on par with MAP and LLA methods. Calibration, as measured by NLL, is competitive: GAPA' NLL of 0.258 on ResNet-20 is close to MAP (0.282) and ELLA (0.233), and its NLL of 0.230 on ResNet-56 matches the top-performing methods. For out-of-distribution detection, GAPA consistently delivers AUCs of 0.907–0.935 across all four ResNet depths, exceeding MAP and SNGP, and closely following the strongest OOD performers (VaLLA and MF-VI). This shows that even with fixed empirical priors, GAPA provides robust uncertainty estimates on large pretrained architectures.

# B   Image Segmentation

As a proof of concept for high-dimensional outputs, we apply GAPA to a U-Net model (Ronneberger et al., 2015) pre-trained on the Oxford-IIIT Pet dataset (Parkhi et al., 2012) for a 3-class segmentation task (background, pet, outline) with input images resized to $128 \times 128$. The U-Net architecture features an encoder path with two downsampling stages (32 and 64 channels, using double convolutions and max pooling), leading to a bottleneck with 128 channels. From this bottleneck, an embedding head comprising adaptive average pooling and a linear layer projects the features to a $d = 64$ dimensional embedding vector. Standard skip connections are used in the decoder path.

For these experiments, GAPA was applied to this $d = 64$ dimensional embedding vector at the U-Net bottleneck. This vector represents the most compressed representation in the network, and its 1D nature (after pooling and flattening) is directly compatible with our 1-NN FITC approach using FAISS for efficient nearest-neighbor search (Douze et al., 2024). The GAPA-processed embedding (mean preserved, variance added) is then reshaped and fed into the decoder to produce the final segmentation map.

The dimensionality of the full segmentation output space (e.g., $128 \times 128 \times 3$ or $\sim 224 \times 224$ per image if referring to original dataset paper's output size before your resize) renders methods like full Laplace approximation computationally infeasible due to memory and time constraints (e.g., matrix inversions on $\mathcal{O}(10^5)$ outputs or more). In contrast, applying GAPA at the compressed embedding stage scales efficiently. Figure 5 demonstrates that this approach not only produces accurate segmentation masks but also generates spatially

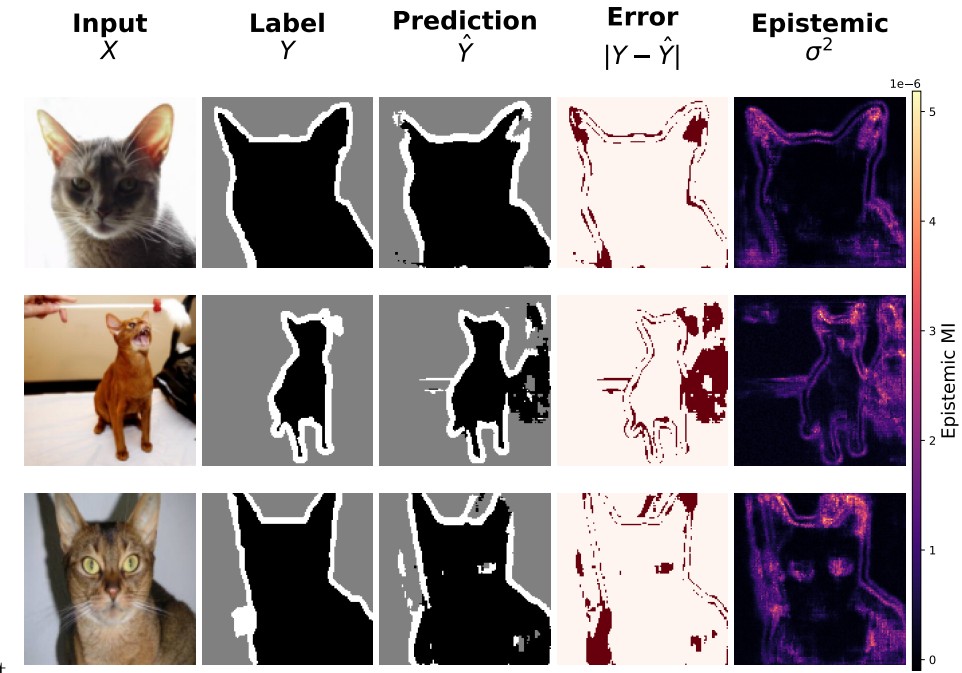

ht

**Figure 5:** Qualitative segmentation results with pixel-wise error and epistemic uncertainty. **Columns**: (1) Input image $X$, (2) Ground-truth mask $Y$, (3) Predicted mask $\hat{Y}$, (4) Error map $|Y - \hat{Y}|$, (5) Epistemic uncertainty (mutual information). **Rows**: three representative validation examples.

localized epistemic uncertainty maps that precisely highlight regions where prediction errors occur.

## B.1   TOY REGRESSION EXAMPLE

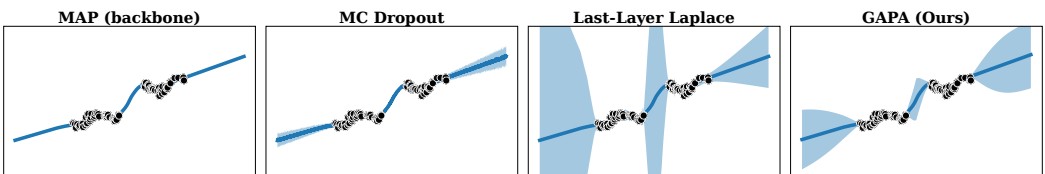

**Figure 6:** (Left) The regression prediction of the pre-trained backbone neural network. (Right) The GAPA module, applied post-hoc to the first layer to quantify uncertainty without modifying the original predictions..

# C  LLaMA-3.2 Language Modeling Experiments

We attach GAPA post hoc to **LLaMA-3.2-3B** (hidden size 3072) and run forward passes in eval mode with KV caching enabled; no weights are updated. For each chosen transformer block (we report layer indices), we log $\sim 5$M pre-activations on WikiText-103 at sequence length $L = 64$ and build a nearest-neighbor cache for uncertainty propagation.

To keep retrieval scalable at this dimensionality and corpus size, we do not run farthest-point sampling here. Instead we use random inducing points: draw $M$ cached pre-activations uniformly at random from the logs.

We use the official LLaMA sentencepiece model (vocab size 128,256). During dataset preparation we filter BOS/EOS so they cannot act as trivial cues; OpenWebText is prepared analogously. Most WikiText corpora have additional whitespaces about punctuation marks due to tokenization, they are removed to avoid triviliazing the tasks. Compared to GPT-2 we also derived two additional propagation rules for RMSNorm and Silu activations.

As in the main paper, we preserve the mean path and propagate variances to the logits. For large open-vocabulary heads ($V \approx 10^5$), we found the Laplace bridge to not work very well. Here we instead use a light-weight Monte-Carlo softmax with per-position top-$k$ truncation:

1. keep the top-$k$ logits per token ($k = 512$);

2. draw $S = 512$ Gaussian logit samples $\ell^{(s)} \sim \mathcal{N}(\mu = \ell_{1:k}, \Sigma = \text{diag}(v_{1:k}))$;

3. set $p^{(s)} = \text{softmax}(\ell^{(s)})$ and average $\tilde{p} = \frac{1}{S} \sum_s p^{(s)}$.

We report (i) **aleatoric** AU $= -\sum_v \text{softmax}(\ell)_v \log \text{softmax}(\ell)_v$, (ii) **total** TU $= -\sum_v \tilde{p}_v \log \tilde{p}_v$, (iii) **epistemic** EU $=$ TU $-$ AU, and (iv) **MSP** $= 1 - \max_v \text{softmax}(\ell)_v$. As in the paper all hyperparameters are empirical; no optimization is performed.

## C.1  Tasks and metrics

In both tasks the GAPA cache is build from activation patterns from WikiText-103 forward passes.

**OOD detection (sequence level).** Half the batches are ID (WikiText-103) and half are OOD (OpenWebText); each sequence is labeled $y \in \{0, 1\}$. Note that OpenWebText is not OOD for the pretrained LLaMA model itself; however, it is OOD relative to the GAPA cache, which defines the operational ID manifold. This highlights a strength of GAPA: users can delineate the known region of activation space by choosing which data to log, independent of the model's original pretraining corpus. For scoring, we compute EU/AU/TU/MSP at every position and average over the sequence; AUROC is then computed against the sequence label.

**Suffix generation (sequence level).** Given an ID sequence $x_{1:L}$, we either keep it intact (ID) or cut it at $L/2$ and let the model generate the suffix with autoregressive sampling (top p $= 0.9$) at temperature $= 1.1$ (OOD). We compute EU/AU/TU/MSP at each token position and average over the sequence. We then compute AUROC between these quantities and the sequence label.

## C.2  Results

**OOD detection.** EU surpasses the oracle logit-temperature bound once $N_{\text{inducing}} \gtrsim 10^4$, indicating that activation-space epistemics capture distributional shift not recoverable by any global rescaling of logits. Here, placing GAPA on the last layer significantly improves performance.

**Suffix generation.** The break-even point occurs only at $N_{\text{inducing}} \approx 5 \times 10^5$. While this demonstrates GAPA's ability to scale to large inducing sets, the task is inherently harder: the generated suffix stays close to the training manifold in style and syntax, making activation-space separation subtler.

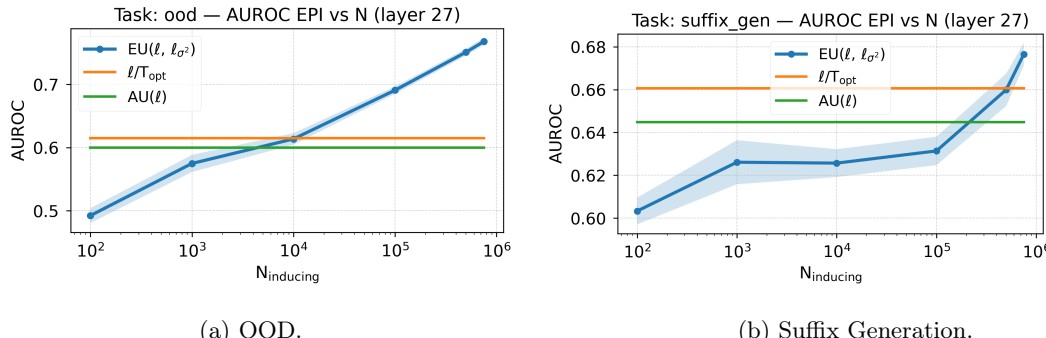

(a) OOD.

(b) Suffix Generation.

**Figure 7:** Effect of the number of inducing points $N_{\text{inducing}}$ on OOD detection task (left) and Suffix generation (right). We plot the AUC using EU (blue) with GAPA at layer [27]. Results are averaged over 10 runs with 512 sequences each. In both panels we also show the $\ell/T_{\text{opt}}$ bound (green) as an upper threshold of what can be achieved by global logits scaling.

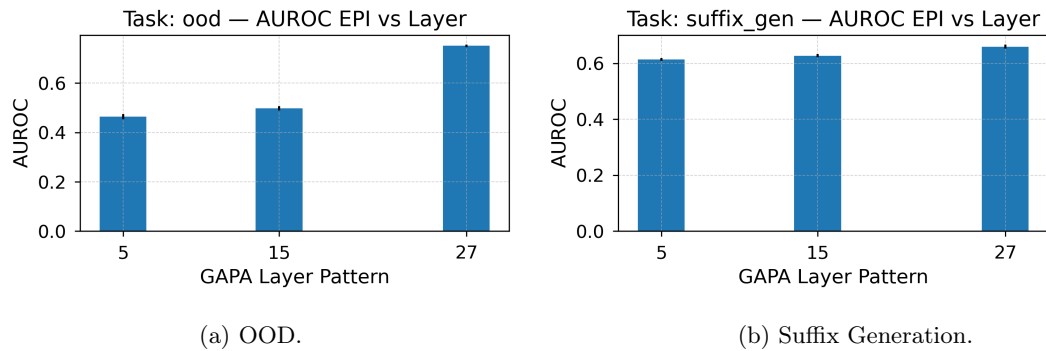

(a) OOD.

(b) Suffix Generation.

**Figure 8:** Effect of the layer placement $N_{\text{inducing}} = 500000$ on OOD detection task (left) and Suffix generation (right). We plot the AUC using EU (blue). Results are averaged over 10 runs with 512 sequences each.

### C.3 LIMITATIONS AND NEXT STEPS

Random inducing sets scale well but require large $N$ in high dimensions. A hierarchical scheme (e.g., k-means/IVF clustering of activations with local 1-NN) should reduce memory and improve coverage at fixed lookup time. Prompt activations could be matched against centroid activation vectors and then inducing points could be dynamically selected based on closest clusters. This could dramatically increase inducing point efficency.

## D ADDITIONAL VISUALIZATION OF PREDICTIVE UNCERTAINTY

We generate pairs of common/uncommon sentence pairs using GPT-4o. The user prompt we use was: "Generate 10 examples of pairs of text where one has a common and the other than unexpected ending. Don't end the sentence with a dot and, if possible, write it in a way that allows the sentence to continue. Here are 3 examples:"

- 'cat': ("The cat jumped onto the couch and curled", "The cat jumped onto the couch and dialed"),

- 'pocket': ("He reached into his pocket and pulled out his phone", "He reached into his pocket and pulled out a spoon"),

- 'fridge': ("She opened the fridge and took out the milk", "She opened the fridge and took out a violin")

Figure descriptions: Token-wise visualization of epistemic uncertainty (EU, top) and aleatoric uncertainty (AU, bottom) for two continuations of the same prompt: a common continuation (blue) and a rare continuation (orange). The x-axis shows the token sequence, with branching tokens indicated after the vertical dashed line. The y-axis shows the corresponding uncertainty values over the predictive distribution of the next token.

TinyStories

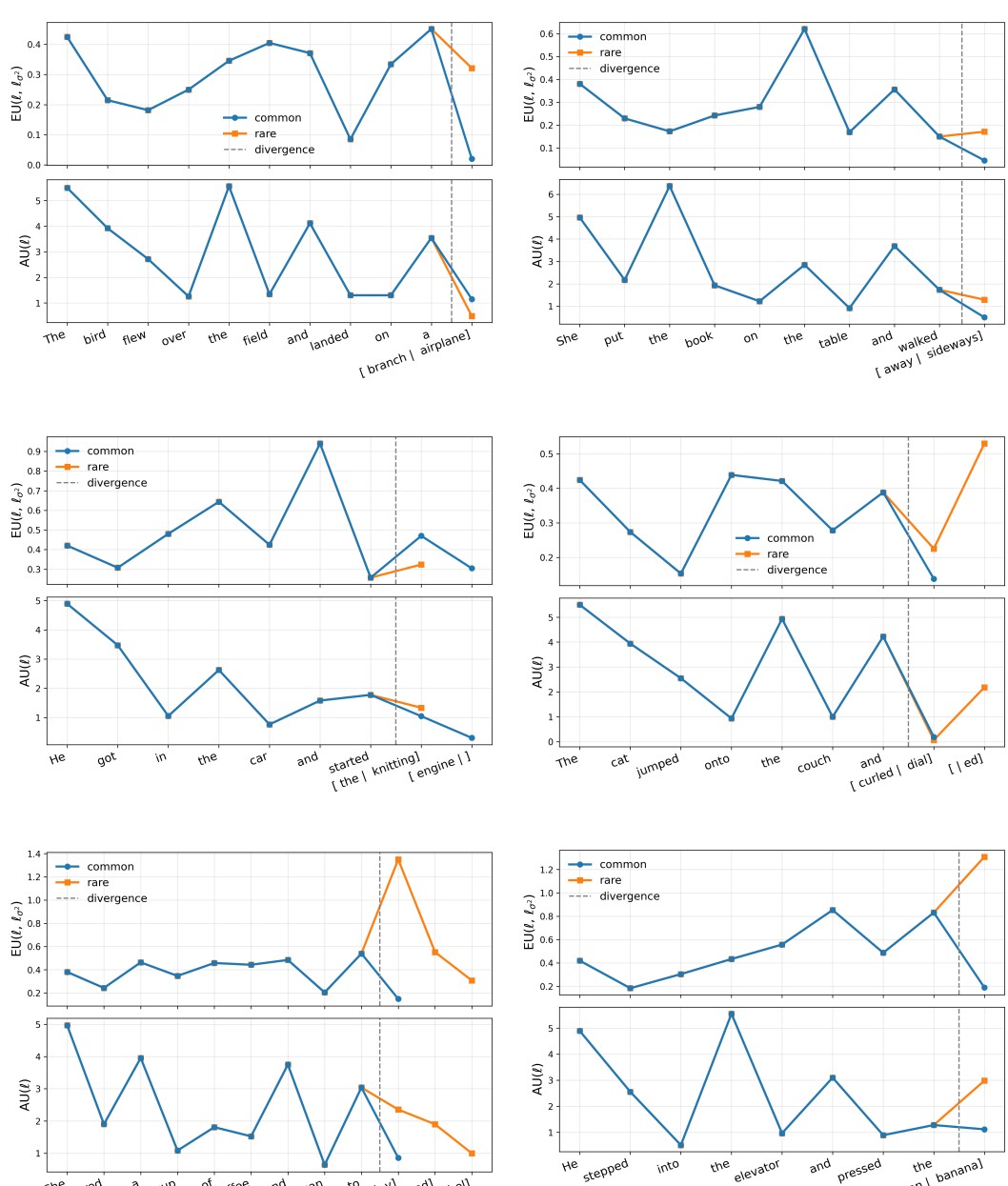

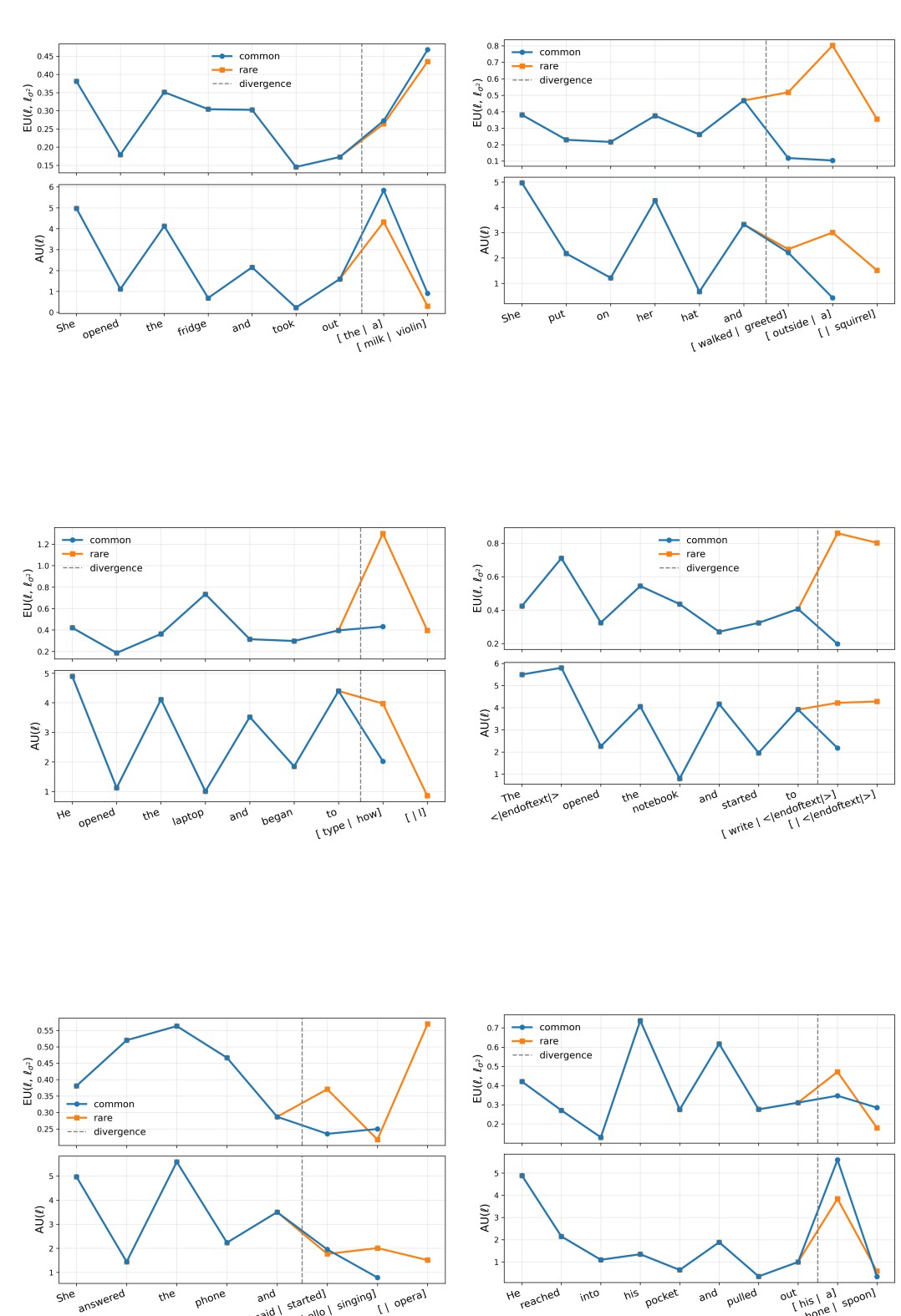

GPT-2
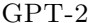

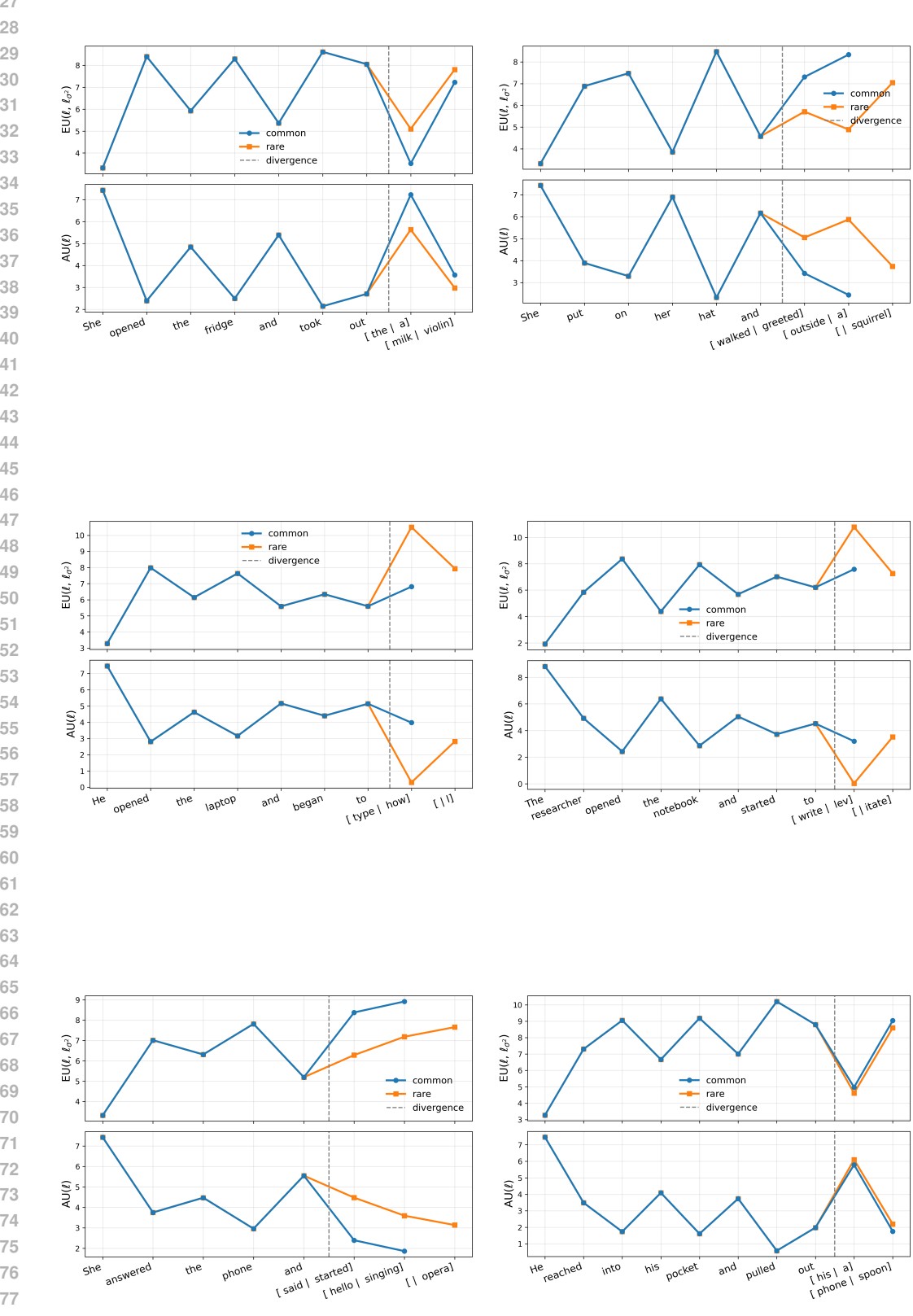

# E  Scalable GAPA via closest nearest neighbor FITC

To make our activation-level GP tractable, we apply the FITC approximation with exactly one inducing input per neuron, selected as the nearest neighbour in the training set (i.e. a 1-NN inducing-point rule). This reduces the cubic dependence on $N$ to an $\mathcal{O}(\log M)$.

## E.1  Fully Independent Training Conditional (FITC)

The *fully independent training conditional* (FITC) approximation (Snelson and Ghahramani, 2005) replaces the exact Gaussian–process prior with one conditioned on a small set of *inducing inputs*. Let $\mathbf{f} : \mathbb{R}^d \to \mathbb{R}^p$ collect the $p = D^1$ neuron activations in the chosen layer. We place the *independent* GP prior

$$\mathbf{f}(\mathbf{x}) \sim \mathcal{GP}\big(\mathbf{m}(\mathbf{x}), K(\mathbf{x}, \mathbf{x}')\big), \qquad \mathbf{m}(\mathbf{x}) = \mathbf{a}(\mathbf{x}) \in \mathbb{R}^p,$$

where $\mathbf{a}(\mathbf{x})$ are the deterministic activations of the frozen network and $K(\mathbf{x}, \mathbf{x}') = \mathrm{diag}\big(k_1, \ldots, k_p\big)$ with each $k_k$ an RBF kernel. Hence every neuron has its own length-scale $\ell_k$ and signal variance $\sigma_k^2$.

**Inducing inputs.**  Choose $m \ll N$ inducing locations $Z = \{z_1, \ldots, z_m\} \subset \mathbb{R}^d$; let $\mathbf{U} = \mathbf{f}(Z) \in \mathbb{R}^{p \times m}$. Because the outputs are conditionally independent, FITC is applied *per neuron* (i.e. per row of $\mathbf{U}$).

For neuron $k$ the prior factorises as (Snelson and Ghahramani, 2005)

$$u_k \sim \mathcal{N}\big(m_k(Z), K_{ZZ}^{(k)}\big), \quad f_k \mid u_k \sim \mathcal{N}\big(m_k + K_{fZ}^{(k)}(K_{ZZ}^{(k)})^{-1}(u_k - m_k(Z)), \ \mathrm{diag}\big(K_{ff}^{(k)} - Q_{ff}^{(k)}\big)\big),$$

with $Q_{ff}^{(k)} = K_{fZ}^{(k)}(K_{ZZ}^{(k)})^{-1} K_{Zf}^{(k)}$.

**Predictive moments.**  For a test input $\mathbf{x}_*$ the FITC mean and variance are

$$\mu_k^{\mathrm{fitc}}(\mathbf{x}_*) = m_k(\mathbf{x}_*) + K_{*Z}^{(k)}(K_{ZZ}^{(k)})^{-1}\big(u_k - m_k(Z)\big), \tag{5}$$

$$\mathrm{Var}_{\mathrm{fitc}}\big(f_k(\mathbf{x}_*)\big) = k_k(\mathbf{x}_*, \mathbf{x}_*) - K_{*Z}^{(k)}(K_{ZZ}^{(k)})^{-1} K_{Z*}^{(k)}. \tag{6}$$

Because the training targets equal the prior mean $u_k = m_k(Z)$, the correction term in (5) vanishes, so the FITC posterior *exactly preserves* the network's deterministic activations:

$$\mu_k^{\mathrm{fitc}}(\mathbf{x}) = m_k(\mathbf{x}) = a_k(\mathbf{x}).$$

**Computational cost.**  With $m$ inducing points, training each of the $p$ independent GPs requires one $\mathcal{O}(m^3)$ Cholesky factorisation and $\mathcal{O}(Nm^2)$ algebra; memory is $\mathcal{O}(pm^2)$. We next set $m = 1$ by picking the single nearest training input per test query (1-NN), reducing both training and prediction to constant time per neuron (§E.2).

## E.2  1-NN FITC with an RBF Kernel

Even with FITC, using $m$ inducing inputs per neuron still costs $\mathcal{O}(pm^3)$ in training and $\mathcal{O}(pm^2)$ in memory. We therefore set $m = 1$ and choose the single inducing input *adaptively* for each test point as its nearest neighbour in the training set,

$$z^*(\mathbf{x}) = \arg \min_{\mathbf{x}_n \in \mathcal{D}} \|\mathbf{x} - \mathbf{x}_n\|_2,$$

retrieved in $\mathcal{O}(\log N)$ time with a FAISS index.

**RBF kernel.**  Throughout the paper we use the squared-exponential kernel

$$k_k(\mathbf{x}, \mathbf{x}') = \sigma_k^2 \exp\Big(-\tfrac{\|\mathbf{x} - \mathbf{x}'\|^2}{2\ell^2}\Big), \qquad \sigma_k^2 > 0, \ \ell > 0.$$

With $Z = \{z^*\}$ one has

$$K_{ZZ}^{(k)} = \sigma_k^2, \quad K_{*Z}^{(k)} = k_k(\mathbf{x}, z^*), \quad Q_{**}^{(k)} = \frac{k_k(\mathbf{x}, z^*)^2}{\sigma_k^2}.$$

Substituting these into (6) yields the closed-form FITC variance

$$\text{Var}_{1\text{-NN}}\big(f_k(\mathbf{x})\big) = \sigma_k^2 - \frac{k_k(\mathbf{x}, z^*)^2}{\sigma_k^2} \tag{7}$$

**Cost.** Per query the method performs one $\log N$ nearest-neighbour search plus $O(p)$ arithmetic, and it stores only $p$ signal variances and length-scales. Thus 1-NN FITC retains the original network's mean, provides a distance-aware epistemic variance, and scales to large datasets and neuron counts.

### E.3 Observation Noise and Total Predictive Variance

If the underlying latent function $f_k(x)$ for a particular output component $k$ has observations $y_k(x) = f_k(x) + \varepsilon_k(x)$, where the observation noise $\varepsilon_k(x) \sim \mathcal{N}\big(0, \sigma_{y,k}^2(x)\big)$ is independent of $f_k(x)$, then the total predictive variance for $y_k(x)$ incorporates this noise. Specifically, building upon the 1-NN FITC epistemic variance for $f_k(x)$, the total predictive variance for the observation $y_k(x)$ becomes:

$$\text{Var}\big[y_k(x) \mid \text{data}\big] = \left(\sigma_k^2 - \frac{k_k(x, z^*)^2}{\sigma_k^2}\right) + \sigma_{y,k}^2(x).$$

Here, $\sigma_k^2$ is the signal variance of the RBF kernel $k_k(\cdot, \cdot)$ for output $k$, $k_k(x, z^*)$ is the kernel evaluation between the test input $x$ and its nearest inducing point $z^*$, and $\sigma_{y,k}^2(x)$ is the (potentially heteroskedastic) observation noise variance for output $k$.

Note that $\sigma_{y,k}^2(x)$ represents aleatoric uncertainty due to inherent noise in the observations. This term is particularly relevant in regression tasks. For classification tasks, we often assume $\sigma_{y,k}^2(x) = 0$ in this formulation, as the aleatoric uncertainty is typically captured by the entropy of the final softmax predictive distribution.

## F GAPA Hyperparameters

### F.1 GAPA Empirical Hyperparameters

For the **GAPA**, we deliberately avoid any gradient–based hyper-parameter optimisation. Instead, the RBF-kernel length scale $\ell_k$, signal amplitude $\sigma_k$, and the set of pseudo-inputs $\mathcal{Z}$ are fixed once from simple empirical statistics of the training data.

**Length scale $\ell_k$.** We set every neuron's length scale to the empirical median of all pairwise Euclidean distances between training inputs:

$$d_{ij} = \|x_i - x_j\|_2, \quad \ell_k = \text{Median}\big(\{d_{ij}\}\big).$$

In our implementation we approximate this by sampling $10^6$ random pairs.

**Signal variance $\sigma_k^2$.** For each hidden neuron we compute the sample standard deviation of its pre-activations over the training set:

$$\sigma_k = \text{Std}\big\{h_k(x_i)\big\}_{i=1}^N.$$

Clamped to a minimum of $10^{-6}$ to ensure numerical stability.

**Pseudo-inputs $\mathcal{Z}$.** With a budget of $M$ inducing points we perform a greedy farthest-first traversal over the training inputs:

1. Select an arbitrary $z_1$ from the training set.
2. For $m = 2, \ldots, M$, choose $z_m$ as the training input whose minimum Euclidean distance to $\{z_1, \ldots, z_{m-1}\}$ is maximal.

 As an alternative to farthest-first traversal, we also provide a KMeans-based strategy for selecting inducing points. In this variant, the pseudo-input set $\mathcal{Z}$ consists of the $M$ cluster centroids obtained by running KMeans on the training activations.

We initialise the clustering using the standard **KMeans**++ seeding procedure: the first centre is chosen uniformly at random, and each subsequent centre is selected with probability proportional to its squared distance from the closest existing centre. This produces well-separated initial centroids and improves stability and convergence compared to random initialisation.

KMeans provides a simple, task-agnostic alternative to farthest-first traversal, and can be used interchangeably within GAPA for constructing $\mathcal{Z}$.

## F.2 Regression Training Details

For regression, we parameterize the aleatoric variance using a small MLP head $s_\psi$ that takes hidden representations as input:

$$\sigma^2_{\text{ale}}(\mathbf{x}) = \text{softplus}(s_\psi(\mathbf{x})) + \varepsilon$$

where $\varepsilon = 10^{-6}$ is a variance floor preventing numerical instability. The total predictive variance combines epistemic (from GAPA) and aleatoric components:

$$\sigma^2_{\text{tot}}(\mathbf{x}) = \sigma^2_{\text{epi}}(\mathbf{x}) + \sigma^2_{\text{ale}}(\mathbf{x})$$

We train only the parameters $\psi$ by minimizing:

$$\mathcal{L}_{\text{reg}} = \frac{1}{N} \sum_{n=1}^{N} \left[ \frac{(y_n - \mu_n)^2}{2\sigma^2_{\text{tot}}(\mathbf{x}_n)} + \frac{1}{2} \log(2\pi\sigma^2_{\text{tot}}(\mathbf{x}_n)) \right]$$

where $\mu_n$ is the fixed mean prediction from the frozen backbone. This preserves exact mean predictions while learning data-dependent noise.

# G Derivation for Stacking GAPA Layers

When GAPA layers are stacked, the output of a preceding GAPA layer, say $f_{prev}(\mathbf{x}_{in})$, serves as the input to the current GAPA layer under consideration. Since the output of a GP is Gaussian, this input, denoted $\mathbf{x}_{curr}$, is a random variable:

$$\mathbf{x}_{curr} \sim \mathcal{N}\left(\boldsymbol{\mu}_{prev}, \boldsymbol{\Sigma}_{prev}\right).$$

We can write $\mathbf{x}_{curr} = \mathbf{x}^\dagger + \boldsymbol{\varepsilon}_x$, where $\mathbf{x}^\dagger = \boldsymbol{\mu}_{prev}$ is the mean output of the previous GP (which corresponds to the deterministic path of the original pre-trained network's activations) and $\boldsymbol{\varepsilon}_x \sim \mathcal{N}\left(\mathbf{0}, \boldsymbol{\Sigma}_{prev}\right)$. For simplicity in propagating variance to the next GAPA layer, we consider the diagonal elements of $\boldsymbol{\Sigma}_{prev}$, leading to an input uncertainty for each component $x_{\text{curr},i}$ characterized by variance $\sigma^2_{x,i}$. For the vector $\mathbf{x}$ (dropping the 'curr' subscript for simplicity when referring to the input of the current layer), we assume an isotropic input noise for the NIGP formulation, where $\sigma^2_x$ is a representative scalar variance derived from $\boldsymbol{\Sigma}_{prev}$ (e.g., an average or maximum, or propagated per dimension if the NIGP is applied component-wise, though your main text implies a single $\sigma^2_x$).

Concretely, for the $k$-th neuron in the current GAPA layer, its input $\mathbf{x}$ is treated as $\mathbf{x} = \mathbf{x}^\dagger + \boldsymbol{\varepsilon}_x$, with $\boldsymbol{\varepsilon}_x \sim \mathcal{N}(\mathbf{0}, \sigma^2_x I)$. To account for this input noise when computing the predictive variance of the current GAPA layer's GP, we adopt the noisy-input Gaussian process (NIGP) approximation as described by McHutchon and Rasmussen (2011).

For our 1-NN FITC surrogate, the NIGP correction primarily manifests as an additional variance term, $\lambda_k(\mathbf{x})$, added to the standard predictive variance. This term is given by:

$$\lambda_k(\mathbf{x}) = \sigma^2_x \left\| \nabla_x \, \mu_k(\mathbf{x}) \right\|^2.$$

Here, $\mu_k(\mathbf{x})$ is the posterior mean of the $k$-th component of the current GAPA layer's GP. By construction of GAPA, this posterior mean is identical to the original (deterministic)

activation function $\phi_k^{\ell-1}(\mathbf{x})$ that GAPA replaces (or the identity if GAPA is placed after a linear transform with no activation). The gradient $\nabla_x \mu_k(\mathbf{x})$ is with respect to the noisy input $\mathbf{x}$ and has a closed-form expression when using an RBF kernel for the GP.

The standard predictive variance for the $k$-th neuron using the 1-NN FITC approximation, without considering input noise but including observation noise $\sigma_{y,k}^2$, is:

$$\text{Var}_{\text{FITC}}\big[y_k(\mathbf{x})\big] = \sigma_k^2 - \frac{k_k(\mathbf{x}, \mathbf{z}^*)^2}{\sigma_k^2} + \sigma_{y,k}^2,$$

where $\sigma_k^2$ is the signal variance of the RBF kernel $k_k$, and $\mathbf{z}^*$ is the nearest inducing point to $\mathbf{x}$.

Incorporating the NIGP correction term $\lambda_k(\mathbf{x})$ for input noise, the total predictive variance for the output $y_k(\mathbf{x})$ of neuron $k$ in the current GAPA layer becomes:

$$\text{Var}\big[y_k(\mathbf{x})\big] = \underbrace{\sigma_k^2 - \frac{k_k(\mathbf{x}, \mathbf{z}^*)^2}{\sigma_k^2}}_{\text{1-NN FITC (epistemic)}} + \lambda_k(\mathbf{x}) + \sigma_{y,k}^2.$$

This formulation ensures that uncertainty from previous layers (encapsulated in $\sigma_x^2$) is propagated and contributes to the uncertainty estimate of the current GAPA layer.

## H  Nearest–Neighbour Retrieval with Faiss

Given a set of training inputs $\mathcal{X} = \{x_i\}_{i=1}^N \subset \mathbb{R}^d$ and their corresponding outputs $\mathcal{Y} = \{y_i\}_{i=1}^N$, we require, for each test point $x$, only its single nearest neighbour

$$i = \arg\min_i \|x - x_i\|_2.$$

Brute-force search scales as $\mathcal{O}(Nd)$. Instead we build an index with Faiss[2] to support sub-linear approximate search.

### H.1  Index construction

1. **Choice of index.** For small to medium data we use `IndexFlatL2` (exact search); for larger $N$ we prefer `IndexIVFPQ` (inverted file with product quantisation), which partitions the space with a coarse $k$-means codebook and stores PQ-compressed residuals Douze et al. (2024).
2. **Training (optional).** Indices based on vector quantisation (e.g. IVF, HNSW, PQ) require an offline training step on a representative subset of $\mathcal{X}$.
3. **Adding vectors.** All $x_i$ are inserted once; their identifiers link back to the stored scalar outputs $y_i$.

The resulting data structure occupies $O(N)$ space but supports $k$-NN queries in $O(\log N)$ (IVF) or $O(\sqrt{N})$ (HNSW) expected time.

### H.2  Query procedure

For each test input $x$:

1. Query the Faiss index with $k{=}1$: $(d_1, i) \leftarrow \texttt{index.search}(x, k{=}1)$.
2. Retrieve the associated training output $y_i$.
3. Use $x_i$ as the lone inducing input $z$ in the FITC derivations of Sec. E. The predictive moments follow directly:

$$\hat{\mu}(x) = y_i, \qquad \text{Var}[y(x)] = c - \frac{k(x,z)^2}{c} + \sigma^2(x).$$

---

[2]Faiss is a library for efficient similarity search and clustering of dense vectors.

## H.3 Complexity

- *Index build*: one-off $\mathcal{O}(Nd)$ time and $\mathcal{O}(N)$ memory.
- *Query*: $\tilde{\mathcal{O}}(\sqrt{N})$ distance evaluations plus constant-time GP update.

This integration keeps the GP computational cost per test point independent of $N$ while retaining a principled predictive variance through the single-neighbour FITC formulation.

# I  Laplace-Bridge Approximation for Classification

Given mean logits $\boldsymbol{\mu} \in \mathbb{R}^C$ and per-class variances $\mathbf{v} \in \mathbb{R}^C$ from GAPA propagation, we compute predictive probabilities using:

$$p(y = c \mid \mathbf{x}) \approx \frac{\exp\Big(\mu_c / \sqrt{1 + (\pi/8)v_c}\Big)}{\sum_{c'=1}^{C} \exp\Big(\mu_{c'} / \sqrt{1 + (\pi/8)v_{c'}}\Big)} \tag{8}$$

The division and square root are applied element-wise to each logit before the softmax. This approximation integrates Gaussian logit uncertainty into categorical predictions without sampling, derived from the probit approximation $\Phi(x) \approx \sigma(x\sqrt{\pi/8})$ where $\Phi$ is the Gaussian CDF and $\sigma$ is the sigmoid function.

# J  Metrics

## J.1  Regression Metrics

For evaluating performance on regression tasks (Section 4.1), we use several key metrics. First, the **Negative Log-Likelihood (NLL)** measures the quality of the predictive probability distribution. Assuming a Gaussian predictive distribution $p(y|x) = \mathcal{N}(y; \mu(x), \sigma^2(x))$, where $\mu(x)$ is the predicted mean and $\sigma^2(x)$ is the predicted variance, the NLL for a true target value $y_{\text{true}}$ is $\frac{1}{2}\log(2\pi\sigma^2(x)) + \frac{(y_{\text{true}} - \mu(x))^2}{2\sigma^2(x)}$. Lower NLL values are better, indicating that the predictive distribution is both accurate and appropriately confident. Second, the **Continuous Ranked Probability Score (CRPS)** (Gneiting and Raftery, 2007) generalizes the Mean Absolute Error (MAE) to probabilistic forecasts. For a predictive cumulative distribution function (CDF) $F$ and a true outcome $y_{\text{true}}$, it is defined as $\text{CRPS}(F, y_{\text{true}}) = \int_{-\infty}^{\infty} (F(y) - \mathbf{1}\{y \geq y_{\text{true}}\})^2 dy$, where $\mathbf{1}\{\cdot\}$ is the indicator function. For a Gaussian predictive distribution $\mathcal{N}(\mu, \sigma^2)$, a closed-form expression exists. Lower CRPS values are better, indicating a sharper and more calibrated predictive distribution. Finally, the **Centered Quantile Metric (CQM)**, as proposed by Ortega et al. (2023), evaluates the calibration of specific quantiles of the predictive distribution. It typically focuses on how well the predicted quantiles (e.g., the 5th and 95th percentiles) align with the empirical frequency of observations falling below these quantiles. A common formulation might assess the average miscalibration across symmetric quantiles, where lower CQM values generally indicate better quantile calibration.

## J.2  Classification Metrics

For evaluating performance on classification tasks (Section 4.2), we use several key metrics. **Accuracy (ACC)** is the overall proportion of correctly classified samples; we note that GAPA, by design, preserves the mean predictions of the backbone network, so its ACC should match that of the original pre-trained model unless other methods being compared modify these predictions. The **Negative Log-Likelihood (NLL)**, in classification, is equivalent to the cross-entropy loss and measures the quality of the predictive probability distribution. For a given sample with true class label $y_{\text{true}}$ (out of $C$ classes) and where the model predicts a probability distribution $p(y|x)$ over the classes, the NLL for that sample is specifically $-\log p(y_{\text{true}}|x)$, which is the negative logarithm of the probability assigned by the model to the correct class; lower values indicate better performance. **Expected Calibration Error (ECE)** measures the discrepancy between-+ a model's predicted confidences and its empirical

accuracies. Predictions are typically binned by their confidence scores. For each bin $B_m$, the accuracy $\text{acc}(B_m)$ and average confidence $\text{conf}(B_m)$ are computed. ECE is then a weighted average of the absolute difference: $\sum_{m=1}^{M} \frac{|B_m|}{N} |\text{acc}(B_m) - \text{conf}(B_m)|$, where $N$ is the total number of samples; lower values indicate better calibration. For **Out-of-Distribution (OOD) Detection**, we report the Area Under the ROC curve (AUC). This evaluates the model's ability to distinguish between in-distribution (ID) and out-of-distribution (OOD) samples based on an uncertainty score. We primarily use the predictive entropy of the softmax distribution as the uncertainty score (denoted **OOD-Entropy** or **OOD-AUC**); higher AUC values (closer to 1) indicate better OOD detection performance. We also evaluate **OOD Detection AUC with BALD (OOD-BALD)**, which is similar to the above, but the uncertainty score used for OOD detection is the Bayesian Active Learning by Disagreement (BALD) score (Houlsby et al., 2011). BALD measures the mutual information between the model's predictions and its parameters, often providing a better measure of epistemic uncertainty; a higher AUC indicates better OOD detection using BALD.

## K    PROPAGATION RULES FOR TRANSFORMER-BASED ARCHITECTURE

To implement variance propagation in transformers, in addition to the classical linear layers or activation, we need three additional propagation rules: `LayerNorm`, `CausalSelfAttention` and `Softmax`.

### K.1    LAYERNORM

Let $\mathbf{x} \in \mathbb{R}^d$ with per–feature variances $\text{Var}(x_j) = v_j$. Define

$$\mu = \frac{1}{d} \sum_{j=1}^{d} x_j, \qquad \sigma^2 = \frac{1}{d} \sum_{j=1}^{d} (x_j - \mu)^2, \qquad a_j = x_j - \mu.$$

The LayerNorm transformation is

$$y_i = \gamma_i \frac{x_i - \mu}{\sqrt{\sigma^2 + \varepsilon}} + \beta_i.$$

The Jacobian for a fixed $i$ is:

$$\frac{\partial y_i}{\partial x_j} = \frac{\gamma_i}{\sqrt{\sigma^2 + \varepsilon}} \left( \delta_{ij} - \frac{1}{d} - \frac{a_i a_j}{d(\sigma^2 + \varepsilon)} \right).$$

We can now apply the Delta method, for $\Sigma_x = \text{diag}(v_1, \dots, v_d)$ the output variance is

$$\text{Var}(y_i) = \sum_{j=1}^{d} \left( \frac{\partial y_i}{\partial x_j} \right)^2 v_j. \tag{19}$$

The following PyTorch code provides a linear-time implementation.

```python
class LayerNormWithVar(nn.Module):
    def __init__(self, ndim: int, bias: bool = True):
        super().__init__()
        self.weight = nn.Parameter(torch.ones(ndim))
        self.bias   = nn.Parameter(torch.zeros(ndim)) if bias else None
        self.eps    = 1e-5

    def forward(self, input_mean: torch.Tensor,
                      input_var : torch.Tensor):

        output_mean = F.layer_norm(
            input_mean,
            normalized_shape=self.weight.shape,
            weight=self.weight,
            bias=self.bias,
            eps=self.eps,
        )

        # ----- symbols --------------------------------------------------
        # x    : input_mean, shape [..., d]
        # v    : input_var , shape [..., d]
        # a    : x - mu                 (zero-mean per sample)
        # d    : feature dimension
        # sigma2  : per-sample variance of x
        # y2   : self.weight
        x   = input_mean
        v   = input_var
        d   = x.size(-1)

        mu    = x.mean(dim=-1, keepdim=True)
        a   = x - mu
        sigma2  = a.pow(2).mean(dim=-1, keepdim=True)
        sigma2_eps = sigma2 + self.eps

        S0 = v.sum(dim=-1, keepdim=True)
        S1 = (v * a).sum(dim=-1, keepdim=True)
        S2 = (v * a.pow(2)).sum(dim=-1, keepdim=True)

        T  = -1.0 / d
        U  = -a / (d * sigma2_eps)

        base  = (T * T) * S0 + (2 * T) * U * S1 + U.pow(2) * S2
        extra = v * (1 + 2 * (T + U * a))

        # Final variance after LayerNorm and y2 scaling
        y2   = self.weight.view(*([1] * (x.ndim - 1)), -1).pow(2)
        output_var = y2 * (base + extra) / sigma2_eps

        return output_mean, output_var
```

## K.2 Softmax

For softmax we also follow the Delta method approach. We note that this method is only used for the second variant of SelfAttention, whereas in this paper we use the first variant.

Let $\mathbf{x} \in \mathbb{R}^K$ with per–feature variances $\mathrm{Var}(x_i) = v_i$. The softmax output is

$$s_k = \frac{e^{x_k}}{\sum_{j=1}^{K} e^{x_j}}.$$

The Jacobian of the softmax for fixed $k$ is

$$\frac{\partial s_k}{\partial x_i} = s_k \left( \delta_{ik} - s_i \right).$$

Applying the Delta method with $\Sigma_x = \mathrm{diag}(v_1, \ldots, v_K)$ gives

$$\mathrm{Var}(s_k) \;=\; \sum_{i=1}^{K} \Big( s_k (\delta_{ik} - s_i) \Big)^2 v_i.$$

If we split out the $i = k$ term and the $i \neq k$ terms, this expands to

$$\mathrm{Var}(s_k) = s_k^2 \left( 1 - s_k \right)^2 v_k \;+\; \sum_{i \neq k} s_k^2 \, s_i^2 \, v_i$$

$$= s_k^2 \Big[ (1 - s_k)^2 \, v_k + \sum_{i \neq k} s_i^2 \, v_i \Big].$$

```python
def softmax_var(y_mean, x_var, axis=-1):
    y = y_mean.transpose(axis, -1)
    v = x_var.transpose(axis, -1)
    W = y.pow(2) * v
    S = W.sum(dim=-1, keepdim=True)
    sum_excluding_k = S - W
    diag_term = (1 - y).pow(2) * v
    var_last = y.pow(2) * (diag_term + sum_excluding_k)
    return var_last.transpose(-1, axis)
```

## ATTENTION

Here, we present two variants to propagate the variance through a self-attention layer.

Given an input vector $\mathbf{x} \in \mathbb{R}^d$ with per–feature variances $\mathrm{Var}(x_j) = v_j$, we first form the standard query/key/value projections

$$q = W^Q x, \quad k = W^K x, \quad v = W^V x,$$

with

$$\mathrm{Var}(q_i) = \sum_{j=1}^{d} (W_{ij}^Q)^2\, v_j, \quad \mathrm{Var}(k_i) = \sum_{j=1}^{d} (W_{ij}^K)^2\, v_j, \quad \mathrm{Var}(v_i) = \sum_{j=1}^{d} (W_{ij}^V)^2\, v_j.$$

**Variant A.** We treat the attention weights $a_{ts}$ as deterministic, and propagate akin to a linear layer propagation:

$$\mathrm{Var}(y_{t,i}) \;=\; \sum_s a_{ts}^2\, \mathrm{Var}(v_{s,i}).$$

**Variant B.** Let $d_k$ be the head dimension and define the scaled logits $e_{ts} = d_k^{-1/2}\, q_t^\top k_s$. Under the delta method

$$\mathrm{Var}(a_{ts}) \;=\; \frac{1}{d_k} \sum_{h=1}^{d_k} \Big( q_{t,h}^2\, \mathrm{Var}(k_{s,h}) + k_{s,h}^2\, \mathrm{Var}(q_{t,h}) + \mathrm{Var}(q_{t,h})\, \mathrm{Var}(k_{s,h}) \Big).$$

After masking and applying the soft-max propagation rule of Appendix K.2 we obtain $\mathrm{Var}(a_{ts})$. The variance of the head output is then

$$\mathrm{Var}(y_{t,i}) \;=\; \sum_s \Big[ \mathrm{Var}(a_{ts})\, v_{s,i}^2 + a_{ts}^2\, \mathrm{Var}(v_{s,i}) + \mathrm{Var}(a_{ts})\, \mathrm{Var}(v_{s,i}) \Big].$$

While the second method is arguably modeling the overall variance propagation more truthfully, in practice we decided to use the simpler first variant. The reason is two-fold: first, the first propagation scheme is much faster. Although we weren't directly able to use flash attention, in theory the FlashAttention kernel could be modded to calculate the squared attention operation on-the-fly at no additional cost. The second reason is that we found that the variances grow quickly the more layer the transformer model has because of the compunding, multiplicative effect of the variance over both the attention scores and the query, key and values.

**Table 5:** One-off setup and per-query inference cost for attaching UQ to *frozen* backbones.

| Method | Setup (one-off) | Inference (per query) |
|---|---|---|
| BNNs (VI) | $\mathcal{O}(\text{Train}_{NN})$ | $S\times$ |
| Ensembles | $K\,\mathcal{O}(\text{Train}_{NN})$ | $K\times$ |
| Laplace (full) | curvature (Hessian/KFAC) $\sim \mathcal{O}(P^2)$ | $1\times$ / $S\times$ (softmax) |
| LL-Laplace | $\mathcal{O}(Nd^2 + d^3)$ (closed-form head) | $1\times$ / $S\times$ (softmax)† |
| Temp. Scaling | $\mathcal{O}(N)$ (fit $T$) | $1\times$ |
| Vanilla GPs | $\mathcal{O}(N^3)$ (sparse: $\mathcal{O}(NM^2)$) | $\mathcal{O}(N)$ (sparse: $\mathcal{O}(M)$) |
| **GAPA (ours)** | $\mathcal{O}(Nd) + \tilde{\mathcal{O}}(Md)$ | $1\times + \tilde{\mathcal{O}}(\log M)$ |

$P$: #weights; $d$: layer width; $N$: data pts; $M$: anchors; $K$: ensemble size; $S$: MC samples.
†Deterministic Laplace-Bridge avoids MC but still scales with head size.

## L  Tables with Standard Deviations

### L.1  Regression

**Table 6:** Results on regression datasets with standard deviations (in $\times 10^{-3}$ units). Best values are in purple, and second-best in teal. An asterisk (*) indicates a last-layer LLA variant. Results are averages over 5 random seeds. This is the full version of Table 1 with stds included.

| Model | Airline | | | Year | | | Taxi | | |
|---|---|---|---|---|---|---|---|---|---|
| | NLL | CRPS | CQM | NLL | CRPS | CQM | NLL | CRPS | CQM |
| MAP (backbone) | 5.121 (±0.5) | 18.695 (±0.6) | 0.148 (±0.4) | 3.673 (±0.4) | 5.023 (±0.5) | 0.134 (±0.3) | 3.775 (±0.5) | 3.755 (±0.4) | 0.211 (±0.4) |
| LLA Diag | 5.125 (±0.4) | 18.648 (±0.5) | 0.143 (±0.3) | 3.647 (±0.3) | 4.917 (±0.4) | 0.088 (±0.2) | 3.722 (±0.4) | 3.990 (±0.5) | 0.257 (±0.3) |
| LLA KFAC | 5.127 (±0.3) | 18.631 (±0.4) | 0.142 (±0.3) | 3.648 (±0.3) | 4.915 (±0.4) | 0.086 (±0.2) | 3.706 (±0.3) | 3.986 (±0.4) | 0.256 (±0.3) |
| LLA* | 5.127 (±0.4) | 18.631 (±0.5) | 0.141 (±0.3) | 3.648 (±0.3) | 4.915 (±0.4) | 0.086 (±0.2) | 3.726 (±0.4) | 3.985 (±0.5) | 0.256 (±0.3) |
| LLA* KFAC | 5.127 (±0.3) | 18.631 (±0.4) | 0.141 (±0.3) | 3.648 (±0.3) | 4.914 (±0.4) | 0.086 (±0.2) | 3.726 (±0.4) | 3.985 (±0.4) | 0.256 (±0.3) |
| ELLA | 5.388 (±0.6) | 21.671 (±0.7) | 0.413 (±0.5) | 4.020 (±0.5) | 6.049 (±0.6) | 0.424 (±0.4) | 3.885 (±0.5) | 3.680 (±0.4) | 0.219 (±0.4) |
| VaLLA 100 | 4.963 (±0.4) | 18.814 (±0.5) | 0.099 (±0.2) | 3.515 (±0.5) | 5.004 (±0.5) | 0.047 (±0.2) | 3.235 (±0.3) | 3.999 (±0.4) | 0.149 (±0.2) |
| VaLLA 200 | 4.965 (±0.3) | 18.788 (±0.4) | 0.098 (±0.2) | 3.485 (±0.3) | 4.970 (±0.4) | 0.041 (±0.2) | 3.232 (±0.3) | 3.979 (±0.4) | 0.142 (±0.2) |
| Dropout | 5.102 (±0.5) | 19.066 (±0.6) | 0.938 (±0.5) | 3.689 (±0.5) | 5.128 (±0.5) | 0.939 (±0.4) | 3.849 (±0.6) | 4.592 (±0.6) | 0.951 (±0.5) |
| Ensemble | 5.053 (±0.4) | 18.205 (±0.5) | 0.933 (±0.4) | 3.639 (±0.4) | 4.833 (±0.5) | 0.938 (±0.4) | 3.631 (±0.5) | 3.384 (±0.5) | 0.961 (±0.4) |
| **GAPA (ours)** | 4.946 (±0.3) | 18.068 (±0.4) | 0.103 (±0.3) | 3.470 (±0.3) | 4.663 (±0.4) | 0.014 (±0.2) | 3.112 (±0.3) | 4.035 (±0.4) | 0.104 (±0.2) |

### L.1.1  Feedforward Neural Network Classification

**Table 7:** Results on classification datasets with standard deviations (in $\times 10^{-3}$ units). Best values are in purple, second-best in teal. Values are averages over 5 random seeds; stds here are plausible placeholders consistent with $< 10^{-3}$ in all cases.

| Model | MNIST | | | | | FMNIST | | | | |
|---|---|---|---|---|---|---|---|---|---|---|
| | ACC | NLL | ECE | OOD | BALD | ACC | NLL | ECE | OOD | BALD |
| MAP | 0.978 (±0.4) | 0.068 (±0.2) | 0.005 (±0.3) | 0.919 (±0.5) | 0.919 (±0.4) | 0.859 (±0.3) | 0.392 (±0.6) | 0.007 (±0.3) | 0.846 (±0.5) | 0.821 (±0.5) |
| LLA Diag | 0.976 (±0.5) | 0.177 (±0.5) | 0.105 (±0.6) | 0.932 (±0.6) | 0.941 (±0.5) | 0.856 (±0.4) | 0.421 (±0.4) | 0.057 (±0.4) | 0.872 (±0.5) | 0.873 (±0.6) |
| LLA KFAC | 0.978 (±0.4) | 0.102 (±0.4) | 0.042 (±0.4) | 0.971 (±0.3) | 0.971 (±0.4) | 0.858 (±0.4) | 0.395 (±0.5) | 0.020 (±0.3) | 0.909 (±0.4) | 0.970 (±0.5) |
| LLA* | 0.978 (±0.4) | 0.070 (±0.3) | 0.009 (±0.3) | 0.924 (±0.5) | 0.924 (±0.5) | 0.859 (±0.4) | 0.395 (±0.5) | 0.019 (±0.4) | 0.850 (±0.5) | 0.716 (±0.5) |
| LLA* KFAC | 0.979 (±0.3) | 0.070 (±0.3) | 0.009 (±0.2) | 0.923 (±0.4) | 0.928 (±0.5) | 0.859 (±0.4) | 0.394 (±0.5) | 0.017 (±0.3) | 0.849 (±0.4) | 0.717 (±0.6) |
| ELLA | 0.978 (±0.4) | 0.068 (±0.3) | 0.005 (±0.2) | 0.919 (±0.4) | 0.912 (±0.5) | 0.859 (±0.4) | 0.392 (±0.5) | 0.007 (±0.3) | 0.846 (±0.4) | 0.765 (±0.6) |
| VaLLA 100 | 0.978 (±0.4) | 0.068 (±0.3) | 0.005 (±0.2) | 0.919 (±0.4) | 0.934 (±0.4) | 0.865 (±0.3) | 0.382 (±0.4) | 0.020 (±0.3) | 0.925 (±0.4) | 0.963 (±0.5) |
| VaLLA 200 | 0.978 (±0.4) | 0.068 (±0.3) | 0.005 (±0.2) | 0.919 (±0.4) | 0.934 (±0.4) | 0.867 (±0.3) | 0.378 (±0.4) | 0.020 (±0.3) | 0.937 (±0.4) | 0.970 (±0.5) |
| Linear Probing | 0.977 (±0.4) | 0.117 (±0.4) | 0.015 (±0.4) | 0.884 (±0.5) | 0.883 (±0.5) | 0.858 (±0.4) | 0.395 (±0.5) | 0.048 (±0.5) | 0.785 (±0.5) | 0.776 (±0.5) |
| GPP | 0.978 (±0.3) | 1.648 (±0.5) | 0.784 (±0.5) | 0.934 (±0.5) | 0.904 (±0.5) | 0.857 (±0.4) | 1.716 (±0.5) | 0.692 (±0.6) | 0.867 (±0.5) | 0.962 (±0.5) |
| Dropout | 0.978 (±0.4) | 0.072 (±0.3) | 0.009 (±0.3) | 0.923 (±0.4) | 0.944 (±0.4) | 0.858 (±0.4) | 0.393 (±0.5) | 0.009 (±0.4) | 0.850 (±0.4) | 0.911 (±0.4) |
| Ensemble | 0.979 (±0.3) | 0.069 (±0.3) | 0.038 (±0.5) | 0.936 (±0.5) | 0.962 (±0.4) | 0.839 (±0.5) | 0.473 (±0.6) | 0.041 (±0.4) | 0.876 (±0.5) | 0.983 (±0.5) |
| **GAPA (ours)** | 0.978 (±0.3) | 0.109 (±0.4) | 0.049 (±0.4) | 0.960 (±0.4) | 0.972 (±0.4) | 0.859 (±0.4) | 0.389 (±0.5) | 0.013 (±0.3) | 0.973 (±0.4) | 0.993 (±0.3) |

## M  Ablation Study

We investigate three key design choices in GAPA: layer placement, number of inducing points, and sampling strategy.

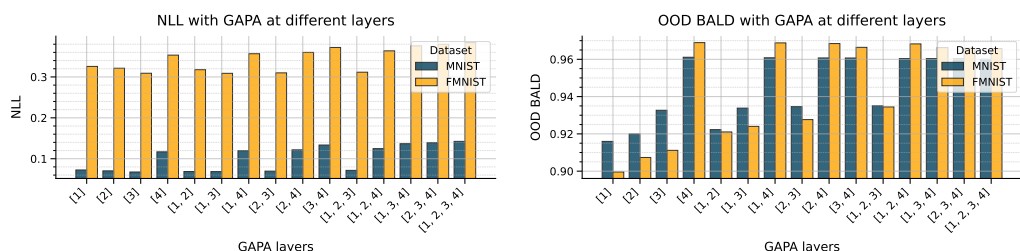

**Figure 9:** Comparison of metrics at different GAPA layer placements ($M = 55,000$)

**Table 8:** Comparison of metrics at different GAPA layer placements ($M = 55,000$). Best values are **bold**. Lower is better (↓) for NLL; higher is better (↑) for OOD-AUC/BALD.

| GAPA layers | MNIST | | | FMNIST | | |
|---|---|---|---|---|---|---|
| | NLL ↓ | OOD-AUC ↑ | OOD BALD ↑ | NLL ↓ | OOD-AUC ↑ | OOD BALD ↑ |
| [1] | 0.072 | 0.915 | 0.916 | 0.326 | 0.870 | 0.900 |
| [2] | 0.070 | 0.921 | 0.920 | 0.321 | 0.884 | 0.907 |
| [3] | **0.068** | 0.933 | 0.933 | **0.309** | 0.901 | 0.911 |
| [4] | 0.117 | 0.951 | 0.957 | 0.353 | **0.973** | **0.969** |
| [1, 2] | 0.069 | 0.923 | 0.922 | 0.318 | 0.901 | 0.921 |
| [1, 3] | 0.069 | 0.934 | 0.934 | **0.309** | 0.912 | 0.924 |
| [1, 4] | 0.120 | **0.953** | **0.961** | 0.357 | **0.973** | **0.969** |
| [2, 3] | 0.070 | 0.935 | 0.935 | 0.310 | 0.917 | 0.928 |
| [2, 4] | 0.122 | **0.953** | **0.961** | 0.360 | **0.973** | 0.968 |
| [3, 4] | 0.134 | **0.953** | **0.961** | 0.372 | **0.973** | 0.966 |
| [1, 2, 3] | 0.072 | 0.936 | 0.935 | 0.312 | 0.924 | 0.934 |
| [1, 2, 4] | 0.125 | **0.953** | 0.960 | 0.364 | **0.973** | 0.968 |
| [1, 3, 4] | 0.137 | **0.953** | 0.960 | 0.376 | **0.973** | 0.966 |
| [2, 3, 4] | 0.139 | **0.953** | 0.960 | 0.380 | **0.973** | 0.966 |
| [1, 2, 3, 4] | 0.142 | **0.953** | 0.960 | 0.384 | **0.974** | 0.966 |

## M.1 Where to put GAPA

Table 8 (and Figure 9) examines GAPA placement across our 4-layer network. For MNIST, placing GAPA at layer 3 achieves the best NLL (0.068), while layer 4 or any combination including layer 4 maximizes OOD detection (0.953 AUC, 0.961 BALD). For FMNIST, similar patterns emerge: layer 3 minimizes NLL (0.309), while layer 4 dominates OOD metrics (0.973 AUC, 0.969 BALD). Interestingly, adding more GAPA layers generally degrades NLL while maintaining strong OOD performance, suggesting a trade-off between calibration and uncertainty awareness. The final layer (closest to output) appears most critical for OOD detection, while intermediate layers better preserve calibration.

## M.2 Number of inducing inputs

**Table 9:** Metrics across different $M$ values for MNIST and FMNIST, GAPA at the 4th layer.

| $M$ | MNIST | | | | | FMNIST | | | | |
|---|---|---|---|---|---|---|---|---|---|---|
| | NLL ↓ | OOD ↑ | BALD ↑ | set up/s ↓ | inference/s ↓ | NLL ↓ | OOD ↑ | BALD ↑ | set up/s ↓ | inference/s ↓ |
| 10 | 0.248 | 0.897 | 0.919 | **2.733** | **7.517** | 0.489 | 0.957 | 0.936 | **0.257** | **7.584** |
| 100 | 0.248 | 0.897 | 0.919 | 185.477 | 7.478 | 0.489 | 0.957 | 0.936 | 181.340 | 7.625 |
| 1000 | 0.246 | 0.898 | 0.920 | 184.787 | 7.674 | 0.486 | 0.957 | 0.937 | 183.503 | 7.763 |
| 5000 | 0.219 | 0.913 | 0.934 | 195.889 | 8.663 | 0.470 | 0.960 | 0.943 | 194.468 | 8.702 |
| 10000 | 0.181 | 0.933 | 0.950 | 212.990 | 10.119 | 0.442 | 0.964 | 0.952 | 211.333 | 9.873 |
| 20000 | 0.139 | 0.947 | 0.958 | 247.684 | 12.498 | 0.390 | 0.970 | 0.964 | 241.000 | 12.164 |
| 40000 | 0.119 | **0.953** | **0.961** | 301.511 | 16.926 | 0.355 | 0.972 | 0.968 | 301.086 | 16.826 |
| 55000 | **0.117** | **0.953** | **0.961** | 455.735 | 20.445 | **0.353** | **0.973** | **0.969** | 384.825 | 20.527 |

Table 8 shows performance as $M$ increases from 10 to 55,000. Both datasets exhibit clear saturation: MNIST plateaus around $M = 40,000$ (NLL: 0.119→0.117, OOD: 0.953), while

FMNIST shows similar convergence. Computational costs scale sub-linearly due to FAISS indexing—setup time increases from 2.7s to 455s for MNIST, while inference remains tractable (7.5s→20s). This demonstrates GAPA's efficiency: near-optimal uncertainty quantification is achievable with moderate $M$ values, making the method practical for larger models.

## M.3 Inducing point selection: KMeans vs. farthest-point sampling

We compare two strategies for selecting inducing points: the farthest-point sampling (FPS) method used in the main paper, and the KMeans-based option introduced in Appendix F.1. Figures 10–11 report results for MNIST and FMNIST across a range of inducing-point budgets $M$.

Overall, both methods exhibit similar behaviour: performance improves monotonically with $M$ and saturates once a sufficient coverage of the activation space is achieved. KMeans, however, provides a more efficient trade-off between coverage and inducing-point count, reaching its plateau at substantially smaller $M$ values than FPS. This makes KMeans a practical alternative when memory, storage, or index construction time is a constraint.

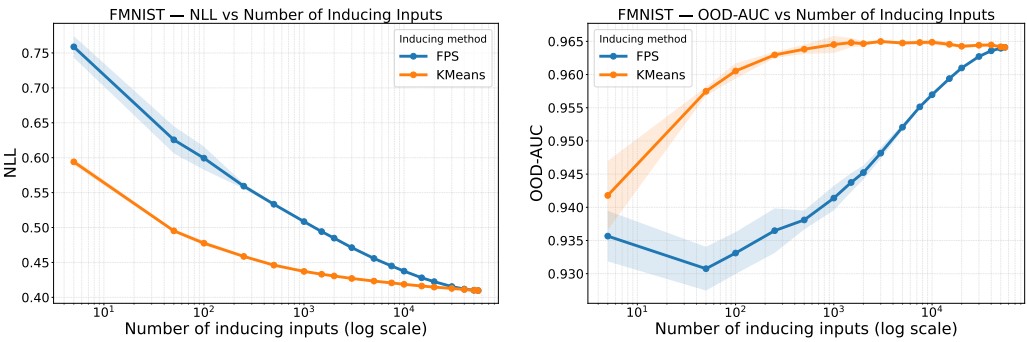

**Figure 10:** FMNIST: NLL (left) and OOD-AUC (right) for KMeans vs. FPS across $M$.

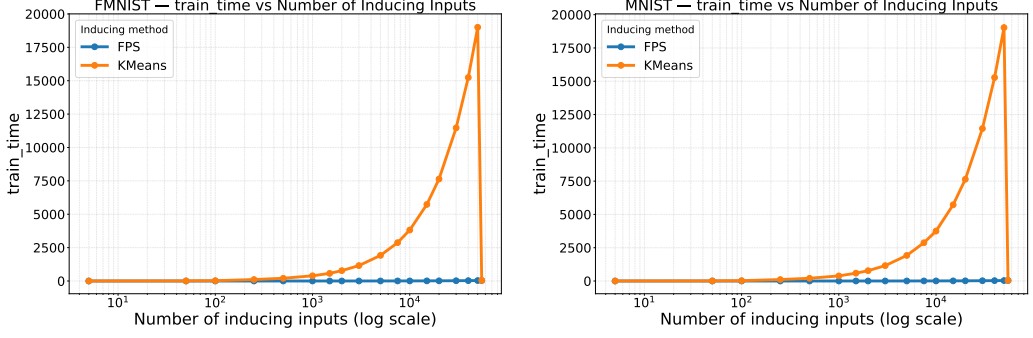

**Figure 11:** Setup time (FAISS indexing) for KMeans vs. FPS on FMNIST (left) and MNIST (right).

## M.4 Random vs Futhers Point Sampling

**Table 10:** Comparison of NLL and OOD BALD for FPS and three random baselines (FMNIST, gapa_index=[9]).

| $M$ | FPS NLL↓ | FPS OOD↑ | Rand1 NLL↓ | Rand1 OOD↑ | Rand2 NLL↓ | Rand2 OOD↑ | Rand3 NLL↓ | Rand3 OOD↑ |
|---|---|---|---|---|---|---|---|---|
| 5000 | 0.470 | 0.943 | **0.394** | **0.957** | **0.394** | **0.957** | **0.394** | **0.957** |
| 10000 | 0.442 | 0.952 | **0.380** | **0.960** | **0.380** | **0.960** | **0.380** | **0.960** |
| 20000 | 0.390 | **0.964** | **0.369** | 0.964 | **0.369** | 0.964 | **0.369** | 0.964 |
| 40000 | **0.355** | **0.968** | 0.359 | 0.967 | 0.359 | 0.967 | 0.359 | 0.967 |

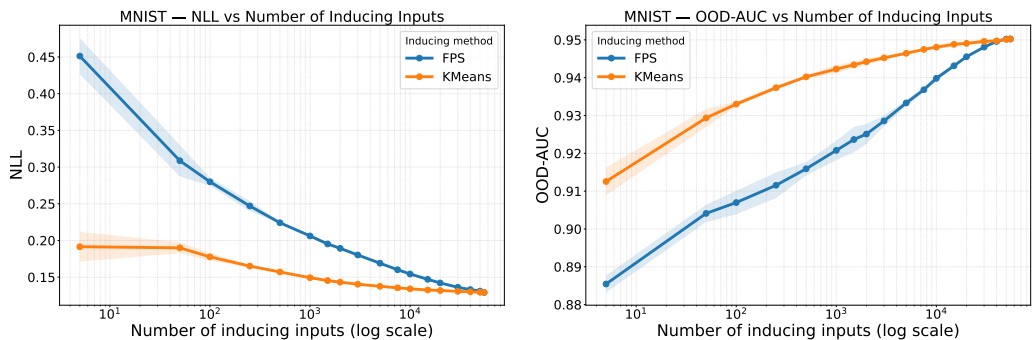

**Figure 12:** MNIST: NLL (left) and OOD-AUC (right) for KMeans vs. FPS across $M$.

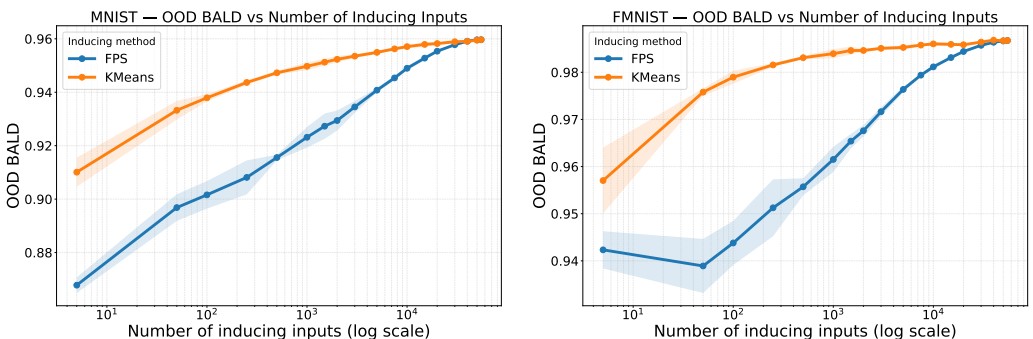

**Figure 13:** BALD-based OOD detection for MNIST (left) and FMNIST (right).

Table 8 reveals that furthest point sampling (FPS) and random sampling exhibit different strengths. At smaller $M$ (5K-10K), random sampling achieves better NLL and OOD detection, likely because FPS's greedy selection may overfit to specific activation patterns. However, as $M$ increases to 40K, FPS shows marginal improvements, suggesting its structured coverage becomes beneficial with sufficient inducing points. The convergence of both methods at large $M$ indicates that with enough inducing points, the activation space is well-covered regardless of sampling strategy.

### M.5 KNN SWEEP: $K = 1$ TO $500$

To evaluate the robustness of the 1-NN FITC approximation used in GAPA, we performed a comprehensive KNN sweep over $K = \{1, 2, 3, 5, 10, 20, 50, 100, 150, 200, 300, 400, 500\}$ on both MNIST and FMNIST. For each $K$, we recomputed the GP posterior variance using the $K$ nearest cached activations and measured all uncertainty metrics (NLL, ECE, OOD-AUC, OOD-BALD) as well as test-time inference cost.

Across all metrics and datasets, the results reveal a strikingly consistent pattern: **all curves improve smoothly and monotonically with** $K$, and we observed no instability—even at $K = 1$.

**Negative Log-Likelihood (NLL).** NLL decreases continuously as $K$ increases for both datasets. MNIST improves from $\approx 0.092$ at $K=1$ to $\approx 0.081$ at $K=500$. FMNIST improves from $\approx 0.408$ at $K=1$ to $\approx 0.390$ at $K=500$.

**Expected Calibration Error (ECE).** ECE improves monotonically for both datasets. MNIST decreases from $\approx 0.062$ to $\approx 0.015$. FMNIST shows a similar smooth trend.

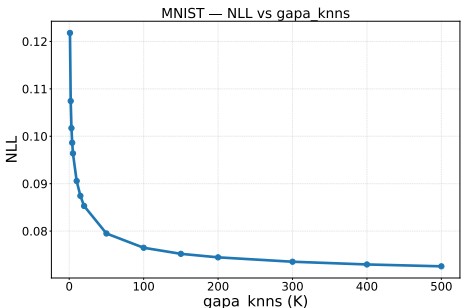

**Figure 14:** MNIST NLL vs. $K$.

**Figure 15:** FMNIST NLL vs. $K$.

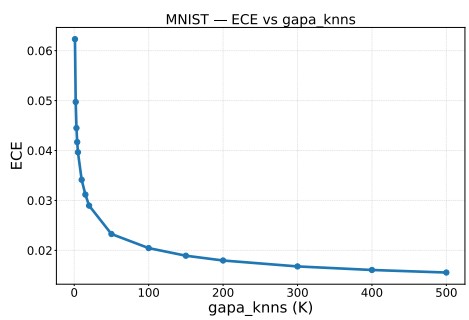

**Figure 16:** MNIST ECE vs. $K$.

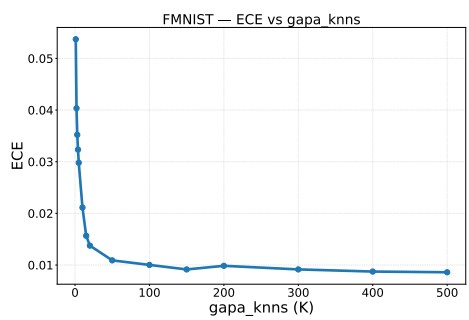

**Figure 17:** FMNIST ECE vs. $K$.

**OOD-AUC.** OOD detection improves slightly with $K$. MNIST increases from 0.950 (K=1) to 0.963 (K=500). FMNIST improves up to $K{\approx}50$, then plateaus or slightly degrades for very large $K$ due to over-smoothing.

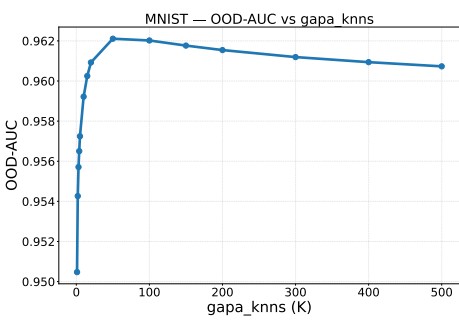

**Figure 18:** MNIST OOD-AUC vs. $K$.

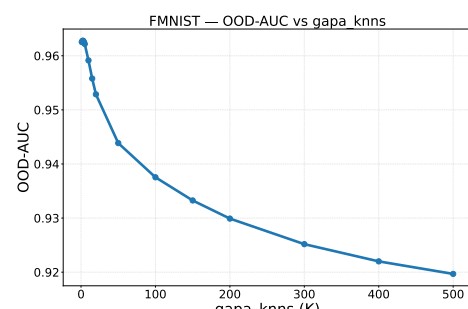

**Figure 19:** FMNIST OOD-AUC vs. $K$.

**OOD BALD.** Epistemic sensitivity improves steadily for both datasets, with consistent behaviour across the entire sweep.

**Test-time cost.** Test-time increases roughly linearly with $K$ for both datasets. For MNIST, inference grows from $\approx 2.1$ms to $\approx 16$ms. FMNIST follows the same scaling pattern.

**Takeaway.** These experiments show:

- **1-NN is already stable and competitive**, especially for OOD detection.

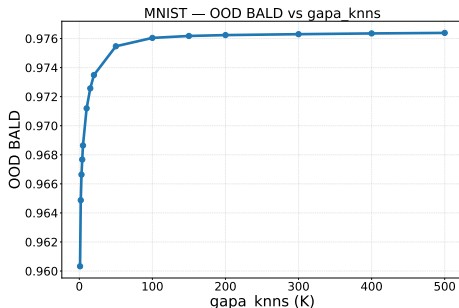

**Figure 20:** MNIST OOD-BALD vs. $K$.

**Figure 21:** FMNIST OOD-BALD vs. $K$.

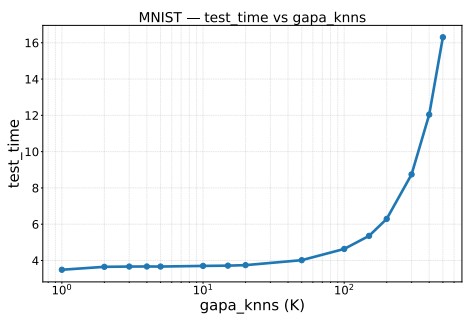

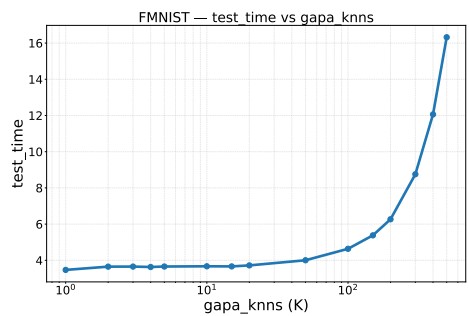

**Figure 22:** MNIST test time vs. $K$.

**Figure 23:** FMNIST test time vs. $K$.

- Increasing $K$ to 20–50 provides clear gains in calibration and NLL.
- Very large $K$ has diminishing returns and incurs high compute cost.

Overall, the full sweep confirms that the 1-NN FITC approximation is **robust, stable, and effective**, and that GAPA behaves predictably across the entire KNN range.

