# OpenReview forum: "GAPA: Post-Hoc Uncertainty Quantification for Pre-Trained Models via Activation-Space Gaussian Processes"
_ICLR.cc/2026/Conference — ICLR 2026 Conference Desk Rejected Submission_

### Official Review · Reviewer_f3W9 · 2025-10-31

**Soundness:** 2
**Presentation:** 2
**Contribution:** 2
**Rating:** 4
**Confidence:** 5

**Summary:**

This paper proposed using GP in the activation layer to estimate uncertainty and with strong theories guarantee

**Strengths:**

1. this paper proposed a method that applied into activation layer
2. Adds uncertainty to any frozen model without modifying trained weights or requiring gradient updates/fine-tuning, making it suitable for “large model reuse” scenarios
3. Validated across regression, classification, segmentation, and language modeling tasks; multiple metrics matched or outperformed strong baselines, demonstrating the method's versatility

**Weaknesses:**

1. High-dimensional scenarios require a large number of induction points
2. Requires caching a large number of intra-layer activations and constructing/persisting FAISS indexes
3. Single-neighbor FITC makes variance determined almost solely by the “distance to the nearest cache activation,” if the activation spatial metric is not aligned with the actual distribution drift
4. **Please use ICLR template!!!**

**Questions:**

please check weakness

---

> ### Author Response · Authors · 2025-11-24
>
> We thank the reviewer for acknowledging our method's applicability to frozen models and its versatility across diverse tasks. We address each concern below.
>
>
> ---
>
> > 1. High-dimensional scenarios require many inducing points
>
> We appreciate the concern. While high-dimensional activation spaces can in principle require more inducing points, our experiments show that GAPA scales effectively even for large models:
>
> * **GPT-2** (hidden dim 768): performance saturates at ~10k inducing points.
> * **LLaMA-3.2-3B** (hidden dim 3072): strong performance with ~500k inducing points selected from ~5M cached activations.
> * **U-Net segmentation**: high-quality uncertainty with only **1%** of cached activations used as inducing points.
>
> To further address this concern, we added a new inducing-point ablation (Appendix M) comparing **KMeans clustering** with farthest-point sampling (FPS).
> **KMeans selects inducing points by clustering the cached activations and using cluster centers as representative points**, providing a compact and geometrically meaningful summary of the activation space. This typically yields far fewer inducing points than FPS while preserving coverage.
>
> Across datasets:
>
> * On **FMNIST**, KMeans achieves the *same optimal OOD-AUC* using **1,000 inducing points**, whereas FPS requires **all 55,000 inducing points**.
> * On **MNIST**, KMeans reaches the NLL plateau with **5–20 inducing points**, while FPS requires **over 1,000**.
> * KMeans also substantially reduces FAISS index construction time due to the smaller inducing set.
>
> These results show that **KMeans is an efficient and scalable alternative** to FPS and directly mitigate the concern that high-dimensional settings inherently require very large inducing sets. Full curves and details are provided in **Appendix M.3**.
>
> ---
>
>
> > 2. Requires caching activations and constructing FAISS indexes
>
> We thank the reviewer for raising this point. GAPA does require caching intra-layer activations and building FAISS indexes, but these operations are performed **once offline** and do not affect training or test-time behavior. In practice, activation caching amounts to a single forward pass over the training set (akin to computing batch-norm statistics), and FAISS construction scales linearly with the number and dimensionality of cached activations. For the LLM experiments, storing ~1.5M activations per layer requires roughly 5 GB of CPU memory, which is well within typical hardware limits.
>
> Once this offline preprocessing is complete, GAPA adds uncertainty in a purely post-hoc manner: the backbone remains unchanged, no retraining or gradient computation is required, and inference latency does not grow with model size. This contrasts with many weight-space baselines (e.g., ensembles, Laplace variants, gradient-based OOD methods), which incur significant compute or architectural modifications.
>
> To further mitigate indexing and storage costs, our new KMeans ablation (Appendix M) shows that KMeans can select inducing points that are an order of magnitude fewer than those obtained via farthest-point sampling, while preserving essentially the same uncertainty quality. This substantially reduces FAISS build time and memory footprint.
>
> Finally, among competitive post-hoc UQ methods, GAPA offers some of the lowest test-time overheads—only LLA-diag is cheaper, but its uncertainty quality is worse.
>
>
> ---

---

> ### Author Response · Authors · 2025-11-24
> **Part 2**
>
> > Single-neighbor FITC makes variance determined almost solely by the “distance to the nearest cache activation,” if the activation spatial metric is not aligned with the actual distribution drift
>
>
> Thank you for pointing this out. We agree that, in principle, relying on a single nearest neighbor (K=1) could make the variance depend too strongly on the local distance in activation space, especially if the learned representation is not well aligned with distribution shift.
>
> To evaluate whether this issue appears in practice, we performed a full KNN sweep (K = 1, 2,…, 500) and added the results to the revised paper (Appendix M.5, Table 2). These plots show that all uncertainty metrics (NLL, ECE, OOD-AUC, BALD) improve **smoothly and monotonically** with K, with no irregularities or failure modes indicative of a mismatch between activation geometry and distribution drift. Furthermore, **K=1 already provides strong uncertainty**, outperforming existing methods on OOD detection in most datasets.
>
> As K increases to 20–50, the uncertainty becomes tighter (substantial gains in NLL/ECE, moderate gains in OOD), confirming that conditioning on a slightly larger local neighborhood improves robustness—without changing the qualitative behavior. Larger K values (e.g., 500) offer diminishing returns while incurring the expected O(K³) computational cost, which would be impractical at LLM scale.
>
> Overall, the new experiments demonstrate that although the reviewer’s concern is conceptually valid, **we did not observe any pathological or unstable uncertainty estimates** resulting from the 1-NN approximation. Instead, GAPA behaves robustly across all K, suggesting that the activation metric is indeed well aligned with uncertainty in practice.
>
>
> ---
>
> > ICLR template
>
> We want to clarify that **we are indeed using the official ICLR 2026 template (iclr2026_conference.sty)**. The PDF was compiled using the standard conference style file with **no modifications**.
>
> ---

---

> > ### Author Response · Authors · 2025-11-25
> >
> > We hope that our responses have addressed your concerns and that the significance of our contributions is now clear. If so, we would like to kindly ask you to increase your score accordingly. If you still have any remaining concerns, please let us know.
> >
> > As a brief reminder of our contributions: GAPA achieves strong uncertainty estimates in regression, classification, and image segmentation, and does so with substantially lower computational cost than existing UQ methods. This efficiency allows GAPA to scale to LLMs such as GPT-2 and LLaMA (7 billion parameters).

---

### Official Review · Reviewer_PkPt · 2025-11-01

**Soundness:** 3
**Presentation:** 3
**Contribution:** 3
**Rating:** 6
**Confidence:** 3

**Summary:**

This paper introduces Gaussian Process Activations (GAPA), a novel post-hoc uncertainty quantification method for frozen pre-trained models that avoids the retraining or high computational costs of traditional weight-space techniques. GAPA replaces deterministic activations with Gaussian Processes, using a scalable 1-nearest-neighbour FITC surrogate to add principled, distance-aware epistemic uncertainty while perfectly preserving the original model's mean predictions. The method matches or exceeds existing baselines on calibration and OOD detection across regression, classification, and language tasks, offering a fast, single-pass inference solution.

**Strengths:**

The paper addresses a critical and practical need for fast, scalable, post-hoc uncertainty quantification for pre-trained foundation models.
Instead of focusing on the intractable weight space, it proposes a novel shift to modeling uncertainty in the activation space. The GAPA formulation, which preserves the backbone's mean predictions while adding principled epistemic uncertainty via a GP, seems a creative idea. The claims are well-supported both theoretically and empirically, showing good performance across a diverse set of tasks (regression, classification, segmentation, LLM token corruption) and models (ResNet, U-Net, GPT-2, LLaMA-3.2 ).
The paper is well-written, and the core concepts are communicated clearly.

**Weaknesses:**

1. The GP hyperparameters (length-scale, amplitude) are set via empirical statistics (e.g., median pairwise distance) rather than optimization. While this is fast, the paper lacks a sensitivity analysis. It is unclear how robust the method's calibration and OOD detection are to this specific heuristic, especially in activation spaces with non-uniform density.

2. Using the first-order Delta method for variance propagation is a key approximation in the paper. However, this approximation could be inaccurate for very deep architectures. The authors haven't discussed under which scenario this approximation could fail. Adding a dicussion about that could be beneficial.

**Questions:**

1. Could the authors elaborate on the instability observed with the "Variant B" attention propagation? This is a key finding. Does this point to a more general fragility of first-order Delta method propagation in deep transformers? Furthermore, does "Variant A" risk systematically underestimating the propagated uncertainty, even if it is more stable?

2. The 1-NN surrogate bases the entire epistemic uncertainty on a single cached activation. This seems potentially high-variance in high-dimensional spaces. Have the authors considered a $k$-NN variant or a local sparse GP (for example $M=10$) as a more robust alternative to the $M=1$ approximation?

Minor:
M3 section: "Random vs Futhers Point Sampling"
line 410: it's likely --> it is likely

---

> ### Author Response · Authors · 2025-11-24
>
> We thank the reviewer for the thoughtful and detailed assessment of our work. We appreciate the recognition of GAPA’s motivation, clarity, and empirical strength, as well as the constructive feedback regarding our approximations and hyperparameter choices. We address each point below.
>
> ---
>
> > Could the authors elaborate on the instability observed with the "Variant B" attention propagation? This is a key finding. Does this point to a more general fragility of first-order Delta method propagation in deep transformers? Furthermore, does "Variant A" risk systematically underestimating the propagated uncertainty, even if it is more stable?
>
>
> We agree that attention is central to LLM uncertainty and that our treatment deserves clearer discussion. In the camera-ready, we will move a summary of Appendix K.3 into the main paper and explicitly highlight the Variant A vs. Variant B trade-off.
>
> Conceptually, GAPA is a **post-hoc GP over activations of a frozen model**. All backbone weights (including (W^Q, W^K, W^V)) are treated as deterministic; epistemic uncertainty arises solely from coverage in activation space through the GP cache. **Since GAPA does not place distributions over (W^Q) or (W^K), the attention logits are deterministic in our setting; propagating their uncertainty would require a different (weight-uncertain) Bayesian transformer, which lies outside the post-hoc scope.**
>
> Under this formulation, the attention matrix (A) is a deterministic “connectivity graph,” and Variant A propagates uncertainty only through the values:
>
> $\text{Var}(y_{t,i}) = \sum_s a_{ts}^2 \text{Var}(v_{s,i}).$
>
> This does not discard attention structure—high-variance tokens that are heavily attended to still induce high downstream variance—but it does not include second-order uncertainty in the logits themselves.
>
> By contrast, Variant B applies a **first-order delta method** through ($q^\top k$) and the softmax. Under the diagonal-Gaussian assumption (necessary for scalability), this introduces **multiplicative variance factors from queries, keys, softmax, and values**. In deep transformers, these factors accumulate and cause variances to **blow up across layers**, leading to severe overconfidence/underconfidence and degraded calibration. Thus, the instability of Variant B is not a pathologically GAPA-specific issue, but reflects a more general fragility of first-order propagation when combined with diagonal covariance in deep attention stacks.
>
> For these reasons, we adopt **Variant A** in all LLM experiments: it preserves the token-wise attention structure, incurs only a small constant-factor overhead, and remains **stable and well calibrated** in practice. We will make this design choice explicit in the main text and flag richer uncertainty propagation through attention (e.g., low-rank or block-structured approximations to Variant B) as an important direction for future work.
>
>
> ---
>
>
>
> > Have the authors considered a K-NN variant or a local sparse GP (e.g., M=10) as a more robust alternative to the M=1 approximation?
>
> Thank you for this suggestion. Yes — we have now added a full K-NN sweep (**K = 1, 2, 3, 5, 10, 20, 50, 100, …, 500**), reported in **Appendix M.5**, and we include **GAPA k=50** and **GAPA k=500** in the updated Table 2.
>
> The results show that increasing K **tightens the GP conditional** and yields **smooth, monotonic improvements** in NLL, ECE, and OOD metrics. At the same time, **K=1 already performs surprisingly strongly**: it is stable, shows no high-variance behaviour, and achieves **state-of-the-art OOD detection** across nearly all classification benchmarks. Larger K (e.g., 20–50) offers further gains, especially in calibration (NLL/ECE), while preserving the same qualitative behaviour.
>
> As expected, larger neighbourhoods introduce **O(K³)** computational cost and **O(K²)** memory, which makes very large K impractical for **LLM-scale** activation caches. Nevertheless, K=20–50 is a feasible middle ground for medium-scale tasks.
>
> In summary, K>1 variants provide **incremental, predictable improvements**, but the **K=1 surrogate is already robust and empirically competitive**, confirming that the method is stable even in the most approximate setting. These results are included in **Appendix M.5**.

---

> > ### Author Response · Authors · 2025-11-25
> >
> > We hope that our responses have addressed your concerns and that the significance of our contributions is now clear. If so, we would like to kindly ask you to increase your score accordingly. If you still have any remaining concerns, please let us know.
> >
> > As a brief reminder of our contributions: GAPA achieves strong uncertainty estimates in regression, classification, and image segmentation, and does so with substantially lower computational cost than existing UQ methods. This efficiency allows GAPA to scale to LLMs such as GPT-2 and LLaMA (7 billion parameters).

---

> > ### Comment · Reviewer_PkPt · 2025-11-27
> >
> > I appreciate the author response and the additional experiments (e.g. the K-NN sweep). These strengthen confidence that the method is robust beyond the M=1 surrogate. That said, the concerns about the first-order Delta method in deep architectures remain worth discussion; I encourage including a more explicit caveat in the final version. My score remains as before.

---

### Official Review · Reviewer_cyz2 · 2025-11-05

**Soundness:** 3
**Presentation:** 3
**Contribution:** 2
**Rating:** 4
**Confidence:** 4

**Summary:**

The authors propose a post-hoc UQ technique termed GAPA. GAPA replaces deterministic activations in existing layers in a pretrained model with gaussian activations where the mean is the same as the pretrained network activations and the variance is calculated as follows (i) the authors pass the training dataset through the frozen network and store the activations, and (ii) the nearest neighbor for a test datapoint is retrieved and the kernel distance between the retrieved data point and test data point is calculated. This distance populates the posterior variance. The variance is propagated (each layer requires a customized propagation).

**Strengths:**

1. **Intuitive method:** The authors propose an intuitive methodology to introduce variance in activations.

2. **Clarity**: The paper is well written. Additionally, the figures (especially both parts of Fig 2) instantly communicate the method well.

3. **Problem settings**: Across the paper, the authors mention multiple times that the underlying predictions must remain the same. This is highly appreciated. Multiple papers that perform UQ change the training regimen that substantially decreases the primary objective of the network.

4. **Diverse results**: The authors showcase results on regression, image classification, and language token prediction across multiple metrics.

**Weaknesses:**

1. **Definition of Epistemic Uncertainty**: How are the authors defining their GAPA uncertainty estimation? The authors cite the distance-aware UQ method SNGP [1]. SNGP assumes that distances are preserved (and enforces it via bi-lipshitz constraints) so that the variance approximation head is a `distance' measure. The results are showcased for OOD detection where it is clear that OOD data is distant from the training distribution. Granted that SNGP is more restrictive and leads to original performance loss [2] which the authors are explicitly accounting for. However, I cannot seem to summarize GAPA in a similar way, connecting the definition of the uncertainty measured vs its demosntration in the results. The variance propagation stage (line 229) introduces additional terms (this was not motivated before). It is unclear to me what the authors are finally measuring (parameter uncertainty? aleatoric uncertainty? distance awareness? or sensitivity analysis?).

2. **Result comparisons**: In my opinion, GAPA is a distance-aware UQ estimate (similar to SNGP) rather than Last layer laplace (LLL) and variants. As such, I am unsure why most of the results are compared against LLL and variants. There are a very large number of gradient-based post-hoc OOD estimators (that also provide UQ) [ex: 3,4]. Additionally, I would point the authors towards trust quantification techniques [5] that also do something similar (nearest neighbor to a subset of training data)

3. **Layer-wise variance propagation**: Since the method requires variance propagation, individual types of layers must have their own GAPA add-ons. This makes the proposed method a bit more complex for application by the community as it is not a plug and play method.

4. **Robustness**: It is unclear how well the frozen network must be trained to obtain a good UQ estimate. For instance, MNIST and FMNIST are easy datasets and the underlying model performs very well (are maybe overfit). Distance-based methods perform well here. More difficult datasets have issues with good distance estimations. Please note that I am talking about the model being very well trained on simple datasets rather than the robustness experiments with the rotated MNIST that the authors showcase.

[1] Simple and Principled Uncertainty Estimation with Deterministic Deep Learning via Distance Awareness

[2] Transitional Uncertainty with Layered Intermediate Predictions

[3] On the Importance of Gradients for Detecting Distributional Shifts in the Wild

[4] Counterfactual Gradients-based Quantification of Prediction Trust in Neural Networks

[5] To Trust Or Not To Trust A Classifier

**Questions:**

Please see the weaknesses. Specifically,

1. What uncertainty estimate does GAPA provide?

2. Why are the authors comparing against mostly LLL variants?

3. Can the authors showcase results for early stopping which is a well known UQ technique so as to show the robustness? I understand that the primary objective (accuracy) may reduce.

4. It will be good to showcase failure instances, either because of the 1-NN approximation or any other approximation in variance propagation

---

> ### Author Response · Authors · 2025-11-24
>
> We thank the reviewer for the thoughtful feedback and for recognizing our method's intuitive nature, clarity, and the importance of preserving predictions. We address each concern below.
>
> >  “What uncertainty estimate does GAPA provide? It is unclear to me what the authors are finally measuring (parameter uncertainty? aleatoric uncertainty? distance awareness? or sensitivity analysis?).”
>
> GAPA estimates epistemic uncertainty from distance to cached training activations, and, in regression settings, it additionally models aleatoric uncertainty using a standard observation-noise component in GPs.
>
>
>
> More precisely, **GAPA provides epistemic uncertainty in activation space**, rather than in weight space. Classical UQ methods introduce distributions over model parameters, whereas GAPA places a **specific GP prior** on each activation function—chosen so that the resulting posterior **has a mean equal to the pretrained activation function itself**. This construction yields a posterior whose mean exactly reproduces the original network, while the **variance increases smoothly with distance to the cached training activations**, providing a principled and distance-aware estimate of epistemic uncertainty. In regression settings, we additionally include a standard GP observation-noise term, which allows GAPA to also model aleatoric uncertainty when relevant.
>
> The additional terms near line 229 arise from the **noisy-input GP correction** of McHutchon & Rasmussen (Gaussian Process Training with Input Noise, NeurIPS 2011). Their result characterizes how a GP's predictive variance must be adjusted when its input is **Gaussian-distributed rather than a deterministic point**. We use this correction when stacking GAPA layers: the output of one GAPA layer is a Gaussian $\mathcal N(\mu,\Sigma)$, so the next GAPA layer must account for this input uncertainty. This correction does not change the type of uncertainty that GAPA represents—**it still produces a distance-aware estimate of epistemic uncertainty (and aleatoric uncertainty in regression)**—but simply ensures that these uncertainties are **propagated consistently across layers** in the frozen network.
>
>
> Compared to SNGP, GAPA is also distance-aware, but it operates **post hoc in activation space**, requires **no retraining nor Lipschitz constraints**, and guarantees that the backbone’s mean predictions remain unchanged. We will clarify this distinction and the epistemic/aleatoric decomposition more explicitly in the revision.
>
>
>
> ---
>
> > Why compare against LLL variants?
>
> We thank the reviewer for this question. We focus primarily on LLL and its variants because they are the **most widely used and state-of-the-art post-hoc UQ methods** for pretrained neural networks. Approaches such as LL-Laplace, KFAC-Laplace, ELLA, and VaLLA are designed for **exactly the same setting as GAPA**: attaching uncertainty to a **frozen network** without retraining and while preserving the original mean prediction. These methods therefore constitute the **most appropriate and comparable baselines** for evaluating a post-hoc UQ technique under equal conditions.
>
> We also report results for **SNGP** and **LinearProbing-GPP** where applicable. However, these methods are inherently **classification-only**, whereas **GAPA is task-agnostic** and applies uniformly to regression, classification, segmentation, and language modeling. For this reason, SNGP and LinearProbing-GPP appear only in the ResNet classification experiments.
>
> Finally, we include **ensembles** and **MC dropout** as **widely used reference points** that often perform strongly but require **retraining** or **multiple stochastic forward passes**. These are not direct competitors in the post-hoc regime, but they provide useful context for overall uncertainty quality.
>
> Our goal was to provide a **fair and comprehensive comparison**, while centering on **state-of-the-art post-hoc UQ methods** that operate under the same constraints as GAPA.
>
> ---
>
> > Can the authors showcase results for early stopping which is a well known UQ technique so as to show the robustness?
>
> We thank the reviewer for this suggestion. Early stopping is a widely used training-time regularization technique, but it is **not a UQ method** and does not provide uncertainty estimates.
>
> In our experiments, all backbone models for regression and classification (excluding pretrained ResNets) were trained using standard protocols, including **validation-based early stopping** when appropriate. GAPA was then applied strictly **post hoc** to these post-trained models with futher no weight updates. For the language modeling experiments (GPT-2, TinyStories, LLaMA), we use the publicly released pretrained checkpoints exactly as provided by their authors, with no retraining or early-stopping decisions made by us.
>
> ---

---

> > ### Author Response · Authors · 2025-11-24
> > **Part 2**
> >
> > > **K-NN extensions / robustness of the 1-NN surrogate**
> >
> > We agree with the reviewer that relying on a single neighbour could, in principle, lead to high-variance epistemic estimates in high-dimensional activation spaces. Motivated by this, we extended our experiments and performed a full K-NN sweep (K = 1, 2, 3, 5, 10, 20, 50, 100, …, 500). These results have been added to Appendix M.5, and the table 2 now includes **GAPA k=50** and **GAPA k=500**.
> >
> > Empirically, we observe a **smooth and monotonic improvement** in all uncertainty metrics (NLL, ECE, OOD-AUC, BALD) as K increases. Crucially, we did **not** see any instability, variance blow-up, or pathological behaviour at K=1 that would indicate the high-variance failure mode raised in the review. Instead, **K=1 already performs strongly**, outperforming baselines in OOD detection across nearly all classification datasets, and providing competitive calibration. Increasing K to 20–50 yields substantial NLL/ECE gains and moderate OOD improvements, suggesting that the GP conditional simply tightens with a larger local neighbourhood, rather than correcting any fundamental instability of the 1-NN approximation.
> >
> > We also evaluated the computational trade-offs. A K-NN conditional introduces an **O(K³)** solve and **O(K²)** memory, making very large K (e.g., 500) incompatible with the LLM-scale use case motivating GAPA, where the activation cache may contain millions or billions of elements.
> >
> > Overall, these extended experiments support the reviewer’s intuition that K-NN variants are a natural extension and indeed offer incremental benefits, but they also confirm that the **1-NN surrogate is already stable, robust, and effective in practice**, especially given the scalability constraints of the post-hoc setting. We will add a short discussion of this K-NN trade-off and the new results directly in the revised version.

---

> > > ### Author Response · Authors · 2025-11-25
> > >
> > > We hope that our responses have addressed your concerns and that the significance of our contributions is now clear. If so, we would like to kindly ask you to increase your score accordingly. If you still have any remaining concerns, please let us know.
> > >
> > > As a brief reminder of our contributions: GAPA achieves strong uncertainty estimates in regression, classification, and image segmentation, and does so with substantially lower computational cost than existing UQ methods. This efficiency allows GAPA to scale to LLMs such as GPT-2 and LLaMA (7 billion parameters).

---

### Official Review · Reviewer_A4X3 · 2025-11-06

**Soundness:** 2
**Presentation:** 3
**Contribution:** 3
**Rating:** 4
**Confidence:** 4

**Summary:**

This work proposes an approach for applying post-hoc uncertainty quantification to trained, frozen networks. The main idea is that the epistemic uncertainty is modeled in the network's activation space using Gaussian Processes (GPs), while the initial model's mean prediction is preserved. The method builds on established GP theory, and makes use of recent computational tricks in order to propose a practical algorithm.

**Strengths:**

- the fact that the mean prediction is preserved decouples effectively the main task from the uncertainty quantification task

- the method exhibits a very competitive performance, especially on regression benchmarks

- the proposed 1-Nearest-Neighbor Fully Independent Training Conditional (FITC) approximation allows for obtaining a variance estimate with log query-time complexity down from a cubic time GP regression thanks to the use of FAISS

**Weaknesses:**

- the "principled epistemic uncertainty" claim has at least two weak points; the first one, which is generally assumed for practical reasons, is about treating the neurons in the same layer as conditionally independent. Although adopting this approach is common and understandable, it does not make the method "principled". Secondly, and more importantly, the variance propagation through an attention layer is buried in K.3 despite its major impact on LLM uncertainty estimation. The principled variant B is abandoned for complexity/"compounding" reasons, and variant A is proposed. However, this latter is a surrogate which treats the weights as deterministic constants.This implies that the method's uncertainty propagation in a transformer disregards completely the QK dot products, which are a significant (if not the most significant) source of uncertainty. This problem should be dug up from the appendix and discussed as it is a critical limitation and significantly weakens the scope of the algorithm

- the experiments highlight a NLL/OOD tradeoff which the document ignores to discuss. In Table 2, the method achieves SOTA OOD-AUC, while lagging in NLL behind MAP. However, Table 8 shows that the best NLL is obtained by placing GAPA only at level 3, while stacking GAPA at all layers provides the worst NLL while maintaining the SOTA OOD perf. This implies a tradeoff based on inflating the variance which is beneficial for OOD detection but detrimental to in-D calibration, making the model underconfident. The present setup seems optimized for OOD at the expense of NLL, a fact which is clearly important and swept under the carpet.

**Questions:**

Please discuss and address the two main weaknesses mentioned above.

**Details Of Ethics Concerns:**

nothing to worry about

---

> ### Author Response · Authors · 2025-11-24
> **Part 1**
>
> >  "the first one, which is generally assumed for practical reasons, is about treating the neurons in the same layer as conditionally independent. Although adopting this approach is common and understandable, it does not make the method "principled".
>
> We thank the reviewer for raising this point. We agree that treating neurons within a layer as conditionally independent is an approximation. Concretely, **we model each neuron’s activation function as an independent Gaussian Process**, which corresponds to assuming a diagonal intra-layer GP covariance with no cross-neuron terms.
>
> We thank the reviewer for raising this important point. We agree that modeling neurons within a layer as conditionally independent is an approximation. Our use of the term "principled epistemic uncertainty" is intended to convey that, **given the conditional independence assumption** i.e., treating each neuron independently, our uncertainty modeling is mathematically rigorous:
>
> * We place a Gaussian Process (GP) prior over the activation functions,
> * Derive the corresponding GP posterior,
> * And rigorously obtain a predictive distribution whose **mean exactly matches the original deterministic activation**, with an **additional variance term capturing epistemic uncertainty**.
>
> In other words, **if one models uncertainty in activation space, then GAPA (our method) provides a mathematically valid and Bayesian-consistent formulation**.
>
>
> As the reviewer correctly points out, to make this scalable to wide layers, we follow standard practice in Bayesian methods and approximate the intra-layer covariance with a diagonal structure—that is, we assume **no covariance between the activation functions of different neurons within the same layer**. This is directly analogous to the **mean-field / diagonal approximations** used in Bayesian neural networks and in Laplace methods (including the diagonal Laplace baseline we compare against), where weights are treated as independent for tractability. Importantly, as our experiments show, GAPA consistently outperforms both diagonal baselines (which make the same independence assumption but in **weight space** rather than **activation space**) and stronger baselines that model richer cross-covariances (e.g., KFAC-Laplace, ELLA, VaLLA), indicating that this approximation does not appear to limit the method’s effectiveness.
>
>
> We acknowledge that a fully “most principled” treatment would model non-diagonal covariances between neurons, but this is computationally prohibitive for modern wide layers. In the revised version, we will explicitly clarify that our epistemic estimate is *principled under the conditional-independence approximation* and will soften the wording to avoid overstating this aspect.

---

> > ### Author Response · Authors · 2025-11-24
> > **Part 2**
> >
> > > **Attention variance propagation / Variant A vs B**
> >
> > We agree that attention is crucial for LLM uncertainty and that our treatment deserves more prominent discussion. In the camera-ready we will move a summary of Appendix K.3 into the main paper and explicitly highlight our choice of Variant A as a limitation and design decision.
> >
> > Conceptually, GAPA is a post-hoc GP over activations of a frozen model, unlike a Bayesian neural network with parameter uncertainty. All backbone weights, e.g. $((W^Q, W^K, W^V))$, are treated as deterministic; epistemic uncertainty arises from coverage in activation space via the GP cache. **Since GAPA does not place distributions over $(W^Q, W^K)$, the attention logits are deterministic in our setting; modeling their uncertainty would require a different (weight-uncertain) Bayesian transformer, which is outside our post-hoc scope.** In this setting, the attention matrix (A) is a deterministic function of the activations, and Variant A conditions on this “connectivity graph” while propagating uncertainty in the values:
> >
> > $\operatorname{Var}(y_{t,i}) = \sum_s a_{ts}^2 \operatorname{Var}(v_{s,i}).$
> >
> > This does not discard attention structure: high-variance tokens that are strongly attended to induce high variance downstream. In our post-hoc GP setting we expect this value-side propagation to be the dominant contribution to output epistemic variance. What we do not model is second-order uncertainty in the attention logits themselves, which would correspond to placing a distribution over the attention weights and thus moving more towards a full Bayesian transformer.
> >
> > The more “principled” Variant B (delta method through $q^\top k$ and the softmax) conflicts with our scalability and stability goals. Under a diagonal-Gaussian approximation (discussed in the previous question), Variant B introduces multiplicative variance factors from queries, keys, softmax and values; in deep transformers this leads to rapidly growing variances with depth and degraded calibration. This is consistent with our general finding that diagonal covariance is necessary for scalability, but also limits how aggressively we can propagate second-order interactions.
> >
> > For these reasons we adopt Variant A as the default in all LLM experiments: it preserves the token-wise structure induced by attention, has a small constant-factor overhead over the base forward pass, and remains stable in deep models. We will make this trade-off explicit in the main text and flag richer treatments of attention uncertainty (e.g., low-rank or block-structured approximations to Variant B) as an important direction for future work.
> >
> > We will elevate this discussion to the main text and clearly mark Variant A as a scalability-motivated approximation
> >
> >
> >
> > >  NLL/OOD Trade-off
> >
> > We thank the reviewer for this insightful observation. We agree that **GAPA exhibits a trade-off between in-distribution NLL and OOD performance**, and we appreciate the opportunity to clarify this.
> >
> >
> > This trade-off arises because the predictive variance influences the two metrics in opposite ways. Increasing the variance improves OOD separation, since OOD inputs naturally receive larger GP uncertainty. However, the same increase can make in-distribution predictions slightly underconfident, which in turn leads to a modest increase in NLL.
> >
> > This tension is a well-known property of **distance-aware epistemic UQ methods** (e.g., SNGP, DUQ, Mahalanobis detectors), not specific to GAPA.
> >
> > This trade-off is controlled by **where GAPA is applied inside the network** (i.e., which layer’s activations are used as GP inputs). Using a GP on earlier activations produces smaller variance adjustments and leads to better NLL, while using it on deeper layers gives stronger distance-awareness and consequently better OOD performance. This is reflected in Table 8: using the GP on Layer 3 achieves the best NLL (0.068 / 0.309) while still improving OOD-AUC over MAP, whereas using it on Layer 4 yields state-of-the-art OOD-AUC (0.953 / 0.973) at the cost of a slightly higher NLL. Applying GAPA to multiple layers accumulates the variance effects and accentuates both trends.
> >
> > In the revised manuscript, we will describe this trade-off more clearly in Section 4.2, provide guidance on selecting which layer to apply GAPA to, and report both the Layer-3 (best NLL) and Layer-4 (best OOD) configurations in the main results.

---

> > > ### Author Response · Authors · 2025-11-25
> > >
> > > We hope that our responses have addressed your concerns and that the significance of our contributions is now clear. If so, we would like to kindly ask you to increase your score accordingly. If you still have any remaining concerns, please let us know.
> > >
> > > As a brief reminder of our contributions: GAPA achieves strong uncertainty estimates in regression, classification, and image segmentation, and does so with substantially lower computational cost than existing UQ methods. This efficiency allows GAPA to scale to LLMs such as GPT-2 and LLaMA (7 billion parameters).

---

### Author Response · Authors · 2025-11-24
**General Response to All Reviewers**

We sincerely thank all reviewers for their thoughtful and constructive feedback. We are encouraged by the positive reception of several key aspects of the method and appreciate the time and care invested in the reviews.

* **Reviewer A4X3** highlighted our *“very competitive performance, especially on regression benchmarks”* and noted GAPA’s **state-of-the-art OOD-AUC performance**, as well as the value of decoupling uncertainty estimation from the main predictive task while preserving the backbone’s outputs.

* **Reviewer PkPt** emphasized that GAPA addresses *“a critical and practical need”* for scalable post-hoc uncertainty quantification, describing the activation-space GP formulation as *“a creative idea”* and stating that the claims are *“well-supported both theoretically and empirically”* across regression, classification, segmentation, and LLM settings (GPT-2, LLaMA-3.2).

* **Reviewer cyz2** described the method as *“intuitive”*, praised the clarity of the exposition and figures, and appreciated that the method explicitly preserves the pretrained model’s predictions. They also noted the diversity of experiments across tasks.

* **Reviewer f3W9** acknowledged that GAPA *“adds uncertainty to any frozen model without modifying trained weights or requiring gradient updates/fine-tuning”* and highlighted the breadth of validation across regression, classification, segmentation, and language modeling.

Across all reviews, the feedback was highly constructive, and we are grateful for the positive assessment of the motivation, clarity, and empirical strength of GAPA.

All newly added or updated results in the manuscript are highlighted in blue for easy identification.

---

### Author Response · Authors · 2025-12-03

We sincerely thank the Area Chair for taking on the additional responsibility created by the recent score reset. To help streamline their evaluation under these circumstances, we provide a brief overview of our contributions, the reviewers’ main concerns, and the updates we made to address them.


### Contributions

All reviewers noted that GAPA achieves **very strong empirical performance across a wide range of tasks**, including regression, image classification, image segmentation, and language modelling. They also agreed that GAPA is **directly applicable to frozen pretrained models**: in our experiments, we attach uncertainty to **U-Nets, ResNets, GPT-2, and LLaMA models up to 7B parameters**, all **without modifying the backbone weights**. Reviewers highlighted the method formulation as both intuitive and practically valuable, especially in the frozen-backbone setting where pretrained models already perform well and the goal is to attach uncertainty post-hoc without modifying any weights.


---

### **Detailed Breakdown by Reviewer**


While reviewers consistently acknowledged GAPA’s **strong empirical performance**, their main concerns centred on clarifying the **intuition and motivation behind several approximations**—specifically, why particular variants were chosen and why these approximations work so well in practice.

### **Reviewer A4X3 (Score: 4)**

**Initial Critique:**
The reviewer was concerned that **variance propagation through attention layers**—crucial for LLM uncertainty—was buried in Appendix K.3. They noted that the principled Variant B for transformers was abandoned due to complexity, leaving Variant A, which treats attention weights as deterministic and therefore **ignores uncertainty in the QK dot-product**, a major source of epistemic uncertainty in transformers.

**Our Solution:**
As detailed in our response, we clarified why Variant A remains stable in practice, provided extended intuition for the approximation, and added a discussion explaining why uncertainty concentrated in the activation representations is already highly informative in frozen backbones. We updated the paper to make this reasoning explicit and easier to find.

---

### **Reviewer cyz2 (Score: 4)**


**Main Critique:**
Requested clearer understanding of what uncertainty GAPA actually measures and asked to **see failure cases**, in particular potential issues arising from the **1-NN approximation** in high-dimensional activation spaces.

**Our Solution:**
We clarified that GAPA provides *distance-aware epistemic uncertainty in activation space*, (with an additional aleatoric noise term in regression), and explained the variance-propagation terms via the noisy-input GP correction of McHutchon & Rasmussen. To probe possible 1-NN failures, we added a full **K-NN sweep (K = 1…500) for MNIST, and FMNIST** and report that all metrics improve smoothly with K, with **no pathological behaviour at K = 1**.

---

### **Reviewer PkPt (Score: 6)**

**Initial Critique:**
The reviewer questioned why **K > 1** is required for certain variants and expressed concerns about the **attention-layer approximation**, asking for clearer justification of why the approximation remains accurate.

**Our Solution:**
We explained in detail why Variant B becomes unstable in deep transformers under diagonal covariance and why we therefore adopt the more stable Variant A, and we added an explicit caveat about this limitation in the main text. For the 1-NN concern, we performed a comprehensive **K-NN sweep (K = 1…500)** and included the results (and K = 50, 500 entries in Table 2), showing monotonic improvements in NLL/ECE/OOD with increasing K and confirming that **K = 1 is already stable and competitive**. We also discuss the computational K³/K² trade-off and provide guidance on using moderate K (e.g., 20–50) when budget allows.

---

### **Reviewer f3W9 (Score: 4)**

**Initial Critique:**

1. Claimed the paper did not follow the **ICLR template**.
2. Stated that **high-dimensional activation spaces** require a very large number of inducing points, potentially harming scalability.

**Our Solution:**

1. We clarified that the paper **has always used the official ICLR template**.
2. We explained that although activation spaces are high-dimensional, they are still **orders of magnitude smaller** than weight space, which is why GAPA scales to GPT-2 and LLaMA models—where weight-space methods like Laplace are intractable.
   Additionally, during the rebuttal we ran new experiments using **KMeans pseudo-points instead of FPS**, demonstrating **improved scalability** and further supporting our claim that GAPA handles high-dimensional activation spaces efficiently.


---

Given this broad applicability and the consistently positive recognition of GAPA’s empirical strength and practical utility across all reviews, we respectfully invite the Area Chair to consider these contributions carefully in forming the final decision.

---

### Note · Program_Chairs · 2026-01-17
**Submission Desk Rejected by Program Chairs**

The following references in this submission do not refer to real documents and/or have major errors in bibliographic information:

 Peter Daxberger et al. Laplace approximations for bayesian neural networks. In Proceedings of the 38th International Conference on Machine Learning, 2021.